# BRCA2 deficiency instigates cGAS-mediated inflammatory signaling and confers sensitivity to tumor necrosis factor-alpha-mediated cytotoxicity

Anne Margriet Heijink[1], Francien Talens[1], Lucas T. Jae[2], Stephanie E. van Gijn[1], Rudolf S.N. Fehrmann[1], Thijn R. Brummelkamp[3,4,5] & Marcel A.T.M. van Vugt[1]

Loss of BRCA2 affects genome stability and is deleterious for cellular survival. Using a genome-wide genetic screen in near-haploid KBM-7 cells, we show that tumor necrosis factor-alpha (TNFα) signaling is a determinant of cell survival upon BRCA2 inactivation. Specifically, inactivation of the TNF receptor (TNFR1) or its downstream effector SAM68 rescues cell death induced by BRCA2 inactivation. BRCA2 inactivation leads to pro-inflammatory cytokine production, including TNFα, and increases sensitivity to TNFα. Enhanced TNFα sensitivity is not restricted to BRCA2 inactivation, as BRCA1 or FANCD2 inactivation, or hydroxyurea treatment also sensitizes cells to TNFα. Mechanistically, BRCA2 inactivation leads to cGAS-positive micronuclei and results in a cell-intrinsic interferon response, as assessed by quantitative mass-spectrometry and gene expression profiling, and requires ASK1 and JNK signaling. Combined, our data reveals that micronuclei induced by loss of BRCA2 instigate a cGAS/STING-mediated interferon response, which encompasses re-wired TNFα signaling and enhances TNFα sensitivity.

---

[1] Department of Medical Oncology, University Medical Center Groningen, Cancer Research Center Groningen, University of Groningen, Hanzeplein 1, 9713GZ Groningen, The Netherlands. [2] Gene Center and Department of Biochemistry, Ludwig-Maximilians-Universität München, Feodor-Lynen-Strasse 25, 81377 Munich, Germany. [3] Oncode Institute, Division of Biochemistry, Netherlands Cancer Institute, Plesmanlaan 121, 1066CX Amsterdam, The Netherlands. [4] Cancer Genomics Center, Plesmanlaan 121, 1066CX Amsterdam, The Netherlands. [5] CeMM Research Center for Molecular Medicine of the Austrian Academy of Sciences, 1090 Vienna, Austria. These authors contributed equally: Anne Margriet Heijink, Francien Talens.  Correspondence and requests for materials should be addressed to M. van Vugt (email: m.vugt@umcg.nl)

C ells are equipped with evolutionary conserved pathways to deal with DNA lesions[1]. These signaling pathways are collectively called the 'DNA damage response' (DDR), and constitute a complex signaling network, displaying multiple levels of cross-talk and feed-back control. Multiple parallel kinase-driven DDR signaling axes ensure rapid responses to DNA lesions, whereas a complementary transcriptional DDR axis warrants maintained signaling. Ultimately, activation of the DDR results in an arrest of ongoing proliferation, which provides time to repair DNA damage. In case of sustained or excessive levels of DNA damage, the DDR can instigate a permanent cell cycle exit (senescence) or initiate programmed cell death (apoptosis)[2].

DNA damage can arise from extracellular sources, including ultraviolet light exposure or anti-cancer treatment, and also originates from intracellular sources, such as oxygen radicals. An alternative source of DNA damage is defective DNA repair. Multiple syndromes are caused by germline mutations in DNA repair genes, which lead to accumulation of DNA damage, and ensuing adverse phenotypes such as accelerated aging, neurodegeneration and predisposition to cancer.

For instance, homozygous hypomorphic mutations of the DNA repair genes BRCA1 and BRCA2 are associated with development of Fanconi anemia[3,4], whereas heterozygous BRCA1 or BRCA2 mutations predispose affected individuals to early-onset breast and ovarian cancer[5-7].

Both BRCA1 and BRCA2 are key players in DNA damage repair through homologous recombination (HR)[8]. BRCA1 functions upstream in HR, where it controls the initiation of DNA-end resection at sites of double-stranded breaks (DSBs), in conjunction with CtIP and the MRN complex[1,2,8]. Once BRCA1 has been recruited to sites of DNA breaks, it associates with PALB2, which ultimately recruits BRCA2. In turn, BRCA2 controls the loading of the RAD51 recombinase onto resected DNA ends[9].

Inactivation of BRCA1, BRCA2 or other HR components severely compromises homology-driven repair of DSBs[8,10,11]. Since HR is vital to repair double-stranded breaks that spontaneously arise during DNA replication, functional HR is required to maintain genomic integrity[9,12-14]. In line with this notion, homozygous loss of Brca1 or Brca2 leads to accumulation of DNA breaks, and results in activation of p53, which promotes cell cycle arrest and activation of apoptosis and senescence programs[15-18]. As a result, BRCA1 or BRCA2 loss is not tolerated during human or mouse development and leads to embryonic lethality[9,12-14]. Importantly, Brca1 or Brca2 are not only essential in the context of development, but also deletion of these genes severely impacts proliferation in vitro, indicating that BRCA1 and BRCA2 are intrinsically essential to cellular viability[12,14,15].

In clear contrast, loss of BRCA1 or BRCA2 is apparently tolerated in breast and ovarian cancers affected by BRCA1 or BRCA2 mutations. It remains incompletely understood how these tumor cells remain viable, despite their continuous accumulation of DNA lesions[19]. The observation that BRCA1 or BRCA2 mutant cancers almost invariably have inactivated TP53 points at p53 signaling forming a barrier to cellular proliferation in the absence of BRCA1 or BRCA2. Indeed, concomitant deletion of Tp53 in mice delays early embryonic lethality in Brca1$^{-/-}$ or Brca2$^{-/-}$ embryos[20,21], and is required to promote tumor formation[22]. However, Tp53 inactivation only partially rescued embryonic lethality and cellular viability of Brca1 or Brca2 mutant cells, indicating that additional mechanisms are likely to play a role in the survival of these cells.

Despite the extensive knowledge of DDR signaling and insight into DNA repair mechanisms, it currently remains incompletely clear how cells with DNA repair defects are eliminated and, conversely, how such cells can escape clearance. Several gene mutations have previously been described to rescue survival of BRCA1-deficient cells, but for BRCA2-deficient cancer cells this remains less clear[23-27]. Here, we used a haploid genomic screen to identify gene mutations that modify cell viability in BRCA2-inactivated cells. We find that loss of the tumor necrosis factor-α (TNFα) receptor, or its downstream signaling component SAM68, rescues cytotoxicity induced by BRCA2 inactivation in KBM-7 cells. Enhanced TNFα appears to be part of a cell-intrinsic and cGAS/STING-dependent interferon response, triggered by formation of micronuclei. Combined, our results describe a mechanism by which autocrine TNFα signaling, induced by cGAS/STING signaling upon loss of the BRCA2 tumor-suppressor gene, limits tumor cell viability.

## Results

**Screening mutations that rescue BRCA2-mediated cell death.** To identify gene mutations that rescue cytotoxicity induced by loss of BRCA2, monoclonal KBM-7 cell lines were engineered to express doxycycline-inducible BRCA2 short hairpin RNAs (shRNAs; Fig. 1a, Supplementary Fig. 1a). To test whether doxycycline treatment resulted in functional inactivation of BRCA2, we tested two previously described functions of BRCA2: facilitating recruitment of RAD51 to sites of DNA breaks[10] and protection of stalled replication forks[28]. After 48 h of doxycycline treatment, ionizing radiation (IR)-induced recruitment of RAD51 to foci was lost (Fig. 1b, Supplementary Fig. 1b). Analogously, the ability to protect stalled replication forks, as assessed by DNA fiber analysis, was weakened significantly (Fig. 1c). Specifically, control cells maintained nascent DNA at replication forks upon hydroxyurea (HU)-induced replication fork stalling. In contrast, BRCA2-depleted cells showed defective protection of stalled forks, as indicated by decreased CldU fiber length after HU treatment (Fig. 1c). Finally, analysis of cell numbers showed that proliferation ceased from 4 days after doxycycline treatment onwards in shBRCA2 cells, and a near-complete loss of cell viability was seen in less than 2 weeks of BRCA2 depletion (Fig. 1d). Importantly, these effects were observed with two independent BRCA2 shRNAs. Notably, KBM-7 cells harbor a loss-of-function TP53 mutation, and our results therefore show that p53 inactivation per se does not preclude the cytotoxic effects of BRCA2 loss[9,20].

The virtually complete cell death after BRCA2 depletion in the near-haploid KBM-7 cells allowed us to use insertional mutagenesis to screen for gene mutations that confer a survival advantage upon BRCA2 depletion (Fig. 1e). To this end, we mutagenized KBM-7-shBRCA2 #2 cells using a retroviral 'gene-trap' vector to obtain a collection of ~100 × 10$^6$ mutants[29,30]. Massive parallel sequencing was performed on genomic DNA isolated from cells that were allowed to grow for 19 days in the presence of doxycycline (Supplementary Data 1). To filter out mutations that affect doxycycline-mediated expression of shRNAs, we performed a cross-comparison with a screen for gene mutations that reversed cell death induced by shRNA-mediated loss of the essential mitotic spindle component Eg5 (Supplementary Data 2)[31]. As expected, multiple dominant integration hotspots identified in the shBRCA2 screen marked doxycycline-related genes which will nullify shRNA-mediated BRCA2 depletion, including SUPT3H, POU2F1 and NONO (Supplementary Data 2 and Fig. 1f). Specifically in BRCA2-depleted cells, we observed an enrichment of insertion sites in the PAXIP1 gene, encoding PTIP, which was recently identified to control replication fork degradation in BRCA2-inactivated cells[27]. Among the most significantly enriched gene mutations, we identified multiple components of the TNFα receptor complex, including TNFRSF1A (encoding TNFR1) and KHDRBS1 (encoding SAM68) (Fig. 1f).

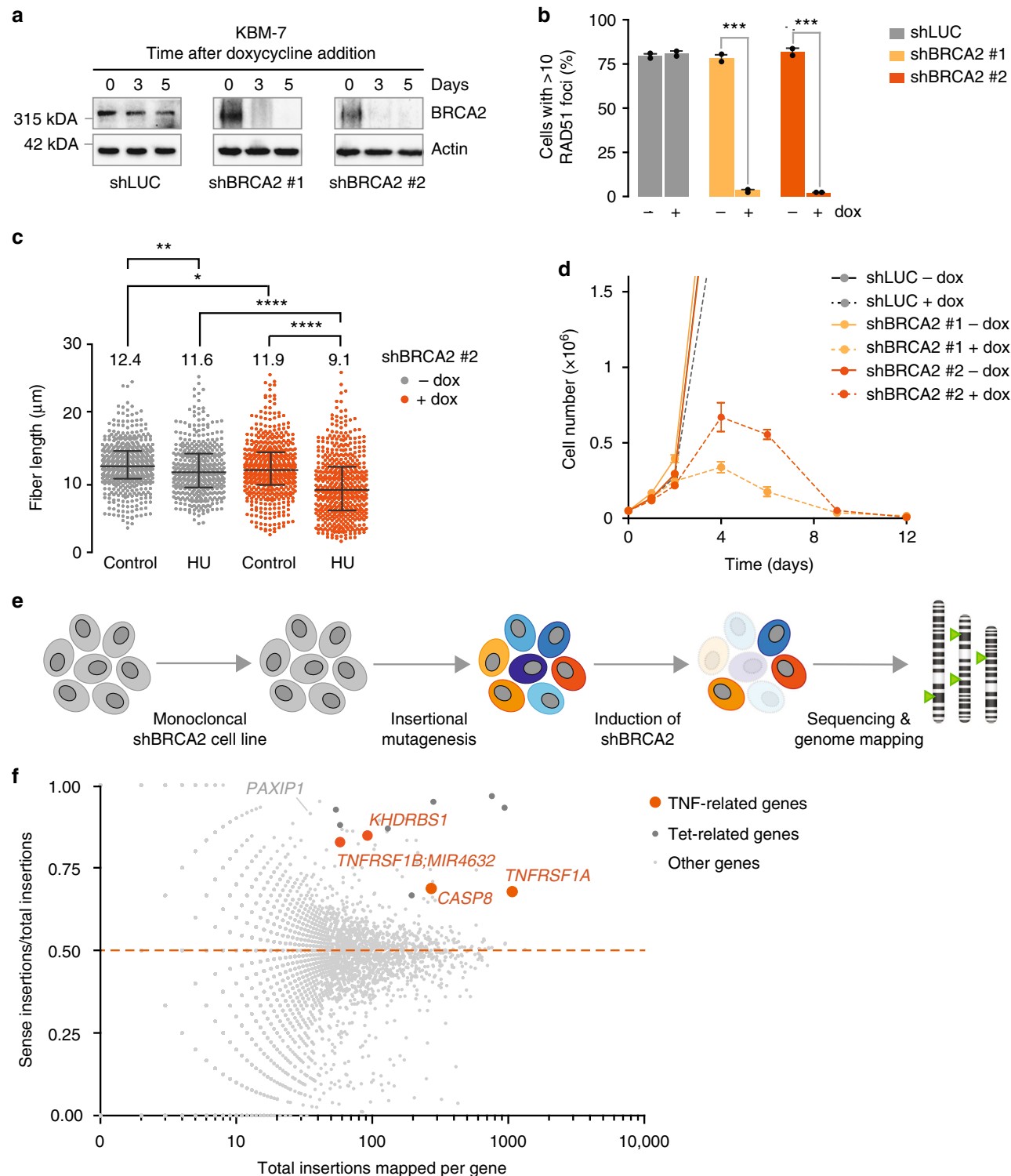

**TNFα signaling determines viability in BRCA2-depleted cells.** To assess whether *BRCA2* mutations in cancers are associated with decreased expression of identified genes, we analyzed the serous ovarian cancers (SOC) within The Cancer Genome Atlas (TCGA) dataset[32]. We specifically analyzed SOC, since *BRCA2* germline mutations are most frequently found within this subgroup of ovarian cancers. Interestingly, the TNFα pathway component *KHDRBS1* on average showed lower median messenger RNA (mRNA) levels in *BRCA2*-mutated vs. *BRCA2* wildtype (wt) tumors (Supplementary Fig. 1c). *KHDRBS1* showed

a larger difference in expression level when compared to *PAXIP1*, although differences for both genes were not statistically significant, likely due to the low number of *BRCA2* mutant cancers. According to literature, the *KHDRBS1* gene product SAM68 is recruited to TNFR1 upon activation with TNFα, where it functions as a scaffold for nuclear factor (NF)-κB activation (Fig. 2a, 'complex 1')[33]. In a delayed response upon TNFα administration, TNFR1 is internalized and SAM68 and RIPK1 disassociate from the TNFα receptor. Together with FADD (Fas-associated protein with death domain) and caspase-8 (Fig. 2a, 'complex 2'), SAM68

**Fig. 1** Genetic determinants of cellular survival in BRCA2-depleted KBM-7 cells. **a** KBM-7 cells were stably transduced with indicated doxycyline-inducible shRNA vectors. Cells were treated with doxycycline for 3 or 5 days and lysates were immunoblotted for BRCA2 and Actin. **b** Quantification of the percentage of cells with ≥10 RAD51 foci after 5 Gy irradiation. KBM-7 cells harboring indicated shRNAs were treated with doxycycline for 96 h prior to irradiation. Approximately 100 cells were scored per condition per replicate. Error bars indicate s.d. of two independent experiments. P values were calculated using two-tailed Student's t-test. **c** KBM-7 cells expressing shBRCA2 #2 were processed for DNA fiber analysis after treatment with doxycycline for 96 h. Cells were then incubated with CldU (25 μM) for 40 min to label replication tracks and subsequently treated with HU (2 mM) for 4 h. CldU track lengths are plotted for ±500 fibers per condition. Median values are indicated and error bars indicate s.d. P values were calculated using two-tailed Student's t-test. **d** Indicated KBM-7 cells were plated in the presence or absence of doxycycline. At indicated time points, cell numbers were assessed. Error bars indicate s.d. of three independent experiments. **e** Workflow of genetic screen in near-haploid KBM-7 cells. **f** Insertions sites identified in gene-trap mutagenized KBM-7 cells which survived doxycycline-induced BRCA2 inactivation (shBRCA2 #2). Dots represent individual genes. The frequency of insertions mapped to a specific gene is plotted on the x-axis. The ratio of gene-traps inserted in the sense orientation over total insertions are plotted on the y-axis. Genes that are neutral in conferring a survival advantage in BRCA2-depleted cells have a sense/total insertion ratio of 0.5 (indicated by the red dashed line). Insertion site ratios > 0.5 represent genes that when mutated confer survival benefit to BRCA2-depleted cells. Throughout the figure, *P < 0.05, **P < 0.01, ***P < 0.001 and ****P < 0.0001

and RIPK1 initiate activation of intrinsic caspases and thereby promote cell death[33].

To validate whether TNFR1 or SAM68 inactivation confers a survival advantage upon BRCA2 depletion, KBM-7-shBRCA2 cells were infected with plasmids harboring shRNAs targeting TNFR1 or SAM68 while also encoding an internal ribosome entry site (IRES)-driven mCherry cassette (Supplementary Fig. 2a). In line with our screening data, BRCA2-depleted KBM-7 cells that were also depleted for TNFR1 or SAM68 showed a survival advantage over cells only depleted for BRCA2, as judged from the gradual increase in mCherry-positive cells (Fig. 2b, c). Notably, TNFR1- or SAM68-depleted KBM-7 cells did not confer a survival advantage through compromising the shRNA-induced knockdown of BRCA2 (Supplementary Fig. 2b). In contrast, BRCA2 depletion was nullified by small interfering RNA (siRNA)-mediated knockdown of SUPTH3, in line with our expectations (Supplementary Fig. 2c).

Depletion of TNFR1 or SAM68 did not reduce the total level of DNA damage induced by BRCA2 loss, as γH2AX levels were similar (Supplementary Fig. 2d). Moreover, loss of TNFR1 or SAM68 did not confer a generic survival advantage, as TNFR1 or SAM68 depletion did not rescue cytotoxicity induced by Eg5 depletion (Supplementary Fig. 2e). It should be noted that TNFα receptor signaling controlled cell death upon BRCA2 loss was not observed in all cell line models. When $Brca2^{F/-}:Tp53^{F/F}$ mouse embryonic fibroblasts (MEFs) were infected with Cre recombinase to induce loss of BRCA2 and p53, this resulted in efficient gene inactivation and interfered with cellular viability (Supplementary Fig. 3a). Of note, shRNA-mediated inactivation of TNFR1 or SAM68 did not significantly rescue cellular survival (Supplementary Fig. 3b,c). In line with these cells not being responsive to TNFα receptor signaling, inactivation of Brca2 did not confer sensitivity to recombinant TNFα (Supplementary Fig. 3d).

Next, two triple negative breast cancer (TNBC) cell lines, BT-549 and MDA-MB-231, were depleted for BRCA2 (Supplementary Fig. 4a). In line with our results in KBM-7 cells, BRCA2 depletion interfered with long-term survival (Supplementary Fig. 4b). Assessing the effects of TNFR1 inactivation on survival of BRCA2-depleted MDA-MB-231 was not feasible, because TNFR1 appeared essential for viability in this cell line, regardless of BRCA2 status (Supplementary Fig. 4c-e). In contrast, a stable population of TNFR1-depleted BT-549 cells was established (Supplementary Fig. 4f,g), and showed that TNFR1 inactivation results in a survival benefit in BRCA2-depleted BT-549 cells (Fig. 2d, e). However, SAM68 depletion interfered with survival of BT-549 cells independent of BRCA2 status, as hardly any cells survived constitutive SAM68 depletion (Supplementary Fig. 4g,h).

Combined, our data show that loss of TNFR1 or SAM68 confers a survival advantage in BRCA2-depleted cells, in situations where TNFR1 or SAM68 are not essential for viability, suggesting a mechanistic link between TNFα signaling and BRCA2 function in a context-dependent fashion.

**Cytokine production and TNFα signaling upon BRCA2 loss.** To further investigate the relation between BRCA2 inactivation and TNFα signaling, we first tested whether limiting the available TNFα pool would alter the reduced cellular viability induced by BRCA2 depletion. Indeed, upon addition of the TNFα-neutralizing antibody infliximab to culture media, cellular viability of BRCA2-depleted KBM-7 cells increased (Fig. 3a). Importantly, and in line with these findings, BRCA2 depletion in KBM-7 cells resulted in increased levels of TNFα secretion, as measured using enzyme-linked immunosorbent assay (ELISA; Fig. 3b). Of note, TNFα production appeared to be part of a broader panel of upregulated pro-inflammatory cytokines in response to BRCA2 depletion, including interleukin (IL)-6 and IL-8 (Fig. 3b). In contrast, the anti-inflammatory cytokine IL-10 was not elevated after BRCA2 depletion (Fig. 3b).

Although the levels of TNFα reproducibly increased upon BRCA2 loss, the overall level of TNFα was limited, and we wondered whether this increase accounted for activation of the TNFα signaling cascade. To test this, we measured the levels of c-Jun N-terminal kinase (JNK) phosphorylation (p-JNK), p38 phosphorylation (p-p38), DNA double-strand break accumulation (γH2AX), and PARP (poly (ADP-ribose) polymerase) cleavage (cPARP) by immunoblotting (Fig. 3c) and flow cytometry (Fig. 3d). Clearly, BRCA2 depletion for 2, 4 or 7 days resulted in increased levels of p-p38 and p-JNK (Fig. 3c, d). In accordance with these observations, the levels of γH2AX and cleaved PARP were also elevated over time upon BRCA2 loss (Fig. 3c–e). Thus, BRCA2 loss instigates a TNFα signaling cascade, leading to cell death in KBM-7 cells, which can be circumvented by sequestering the levels of circulating TNFα.

**BRCA2 inactivation leads to increased TNFα sensitivity.** Since the overall levels of secreted TNFα after BRCA2 depletion were limited, we wondered whether increased cellular sensitivity to TNFα could also play a role. To test whether BRCA2 inactivation increases sensitivity to TNFα, BRCA2-depleted KBM-7 cells were treated with recombinant TNFα. Indeed, BRCA2-depleted but not control-depleted KBM-7 cells showed significantly increased sensitivity to recombinant TNFα (Fig. 4a). Notably, co-depletion of BRCA2 with SAM68 or TNFR1 rescued the observed sensitivity to TNFα (Fig. 4b, Supplementary Fig. 5a). These responses were not specific

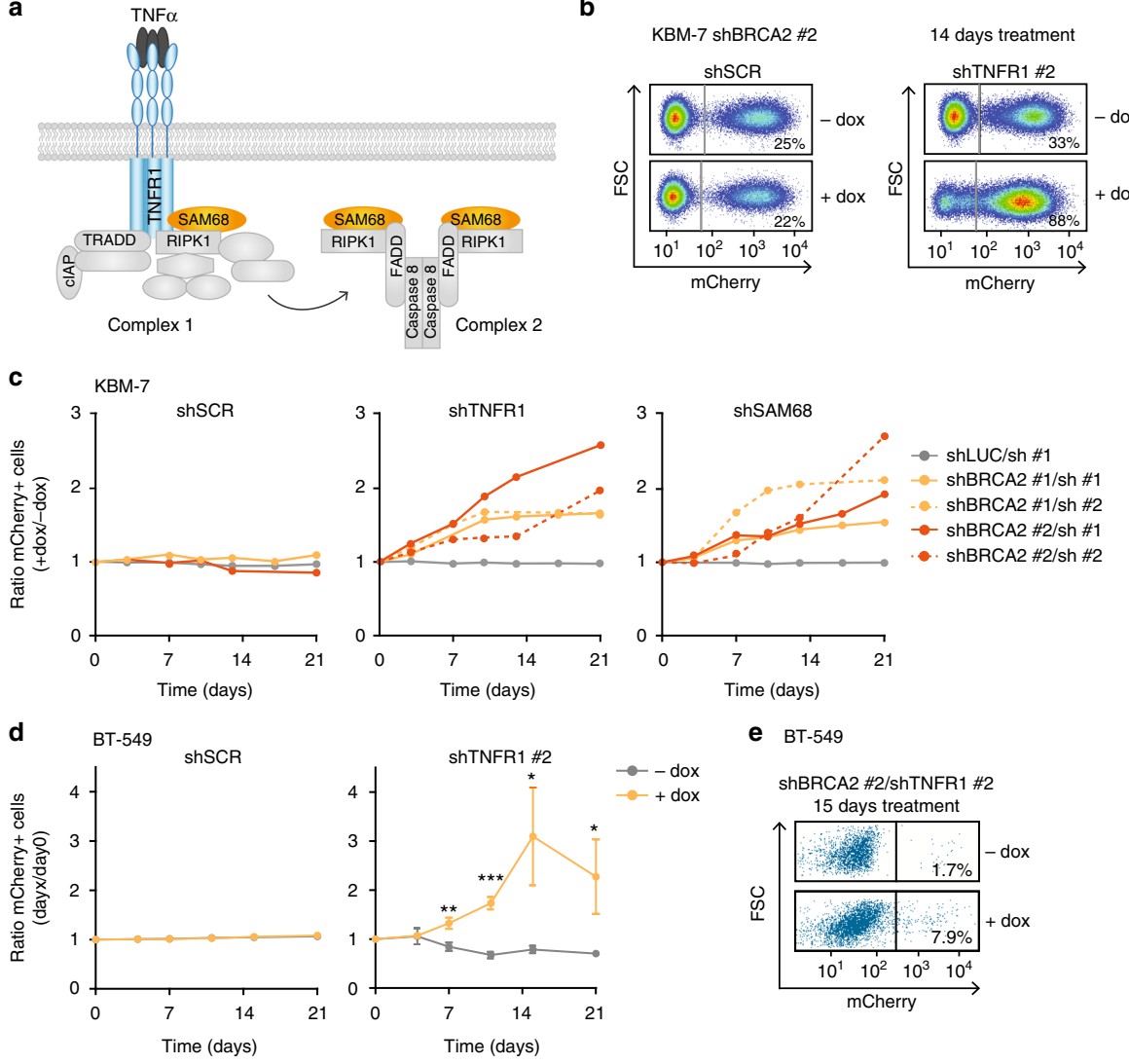

**Fig. 2** Loss of TNFR1 and SAM68 rescues cellular viability in BRCA2-depleted cancer cells. **a** Schematic overview of TNFR1 complex formation upon TNFα binding, leading to cell survival (complex I) or delayed caspase activation and cell death (complex II). **b** Flow cytometry analysis of KBM-7-shBRCA2 #2 cells, additionally carrying indicated shRNA vectors with IRES-driven mCherry cassettes. Cells were treated with doxycycline for 14 days and percentages of mCherry-positive cells were measured. **c** Indicated KBM-7-shBRCA2 cells carrying mCherry shRNA cassettes targeting TNFR1, SAM68 or a control sequence ('SCR') were treated with or without doxycycline to induce BRCA2 shRNA expression. Percentages of mCherry-positive cells were measured every 3 or 4 days for 3 weeks after start of doxycycline treatment. Ratios of mCherry-positive cells in doxycycline treated cultures vs. untreated cultures are indicated. Per condition, at least 30,000 events were measured. **d** BT-549 cells, stably transduced with pLKO.tet.shBRCA2 #2, were infected with IRES mCherry shRNA vectors as for **b**. Cells were treated with or without doxycycline, and percentages of mCherry-positive cells were measured. Ratios of mCherry-positive cells at indicated time points vs. mCherry-positive percentages at day 0 are indicated. Error bars indicate s.d. of three independent experiments. *P* values were calculated using two-tailed Student's *t*-test. *$P < 0.05$, **$P < 0.01$ and ***$P < 0.001$. **e** Representative flow cytometry plots of BT-549-shBRCA2 #2 cells from **d** are shown, carrying mCherry shRNA cassette for TNFR1 #2. Cells were treated for 15 days with or without doxycycline and gated based on mCherry positivity. Numbers indicate the percentages of mCherry-positive cells

to KBM-7 cells, as increased TNFα sensitivity was also observed in a dose-dependent manner upon BRCA2 depletion in a panel of TNBC cell lines (Fig. 4c, Supplementary Fig.4a, 5b) and in the colorectal cancer cell line DLD-1, in which the *BRCA2* gene was inactivated using CRISPR/Cas9 (clustered regularly interspaced short palindromic repeats/CRISPR-associated 9) (Fig. 4d, Supplementary Fig. 5c). Importantly, the increased sensitivity to TNFα in BRCA2-depleted cells could not be attributed to changes in TNFR1 expression levels upon BRCA2 inactivation (Supplementary Fig. 5d, e). Taken together, these results show that BRCA2 inactivation not only induces TNFα signaling, but also results in increased TNFα sensitivity.

In physiological conditions, TNFα-induced NF-κB pro-survival signaling dominates apoptosis signaling[34]. However, sustained activity of JNK (MAPK8), which together with the ASK1 kinase (MAP3K5) acts downstream of the TNF receptor, can 'overrule' NF-κB pro-survival signaling and drive apoptosis[35]. Our observation that BRCA2 depletion leads to increased JNK activity (Fig. 3c, d) would be in line with such a mechanism. To test if sustained activity of ASK1 or JNK kinases is required to mediate TNFα-induced cell death in BRCA2-depleted cells, we chemically inhibited JNK (Fig. 4e, Supplementary Fig. 5f) or ASK1 (Fig. 4f, Supplementary Fig. 5f), in combination with TNFα treatment. Control-depleted BT-549 and HCC38 cells were not sensitive to

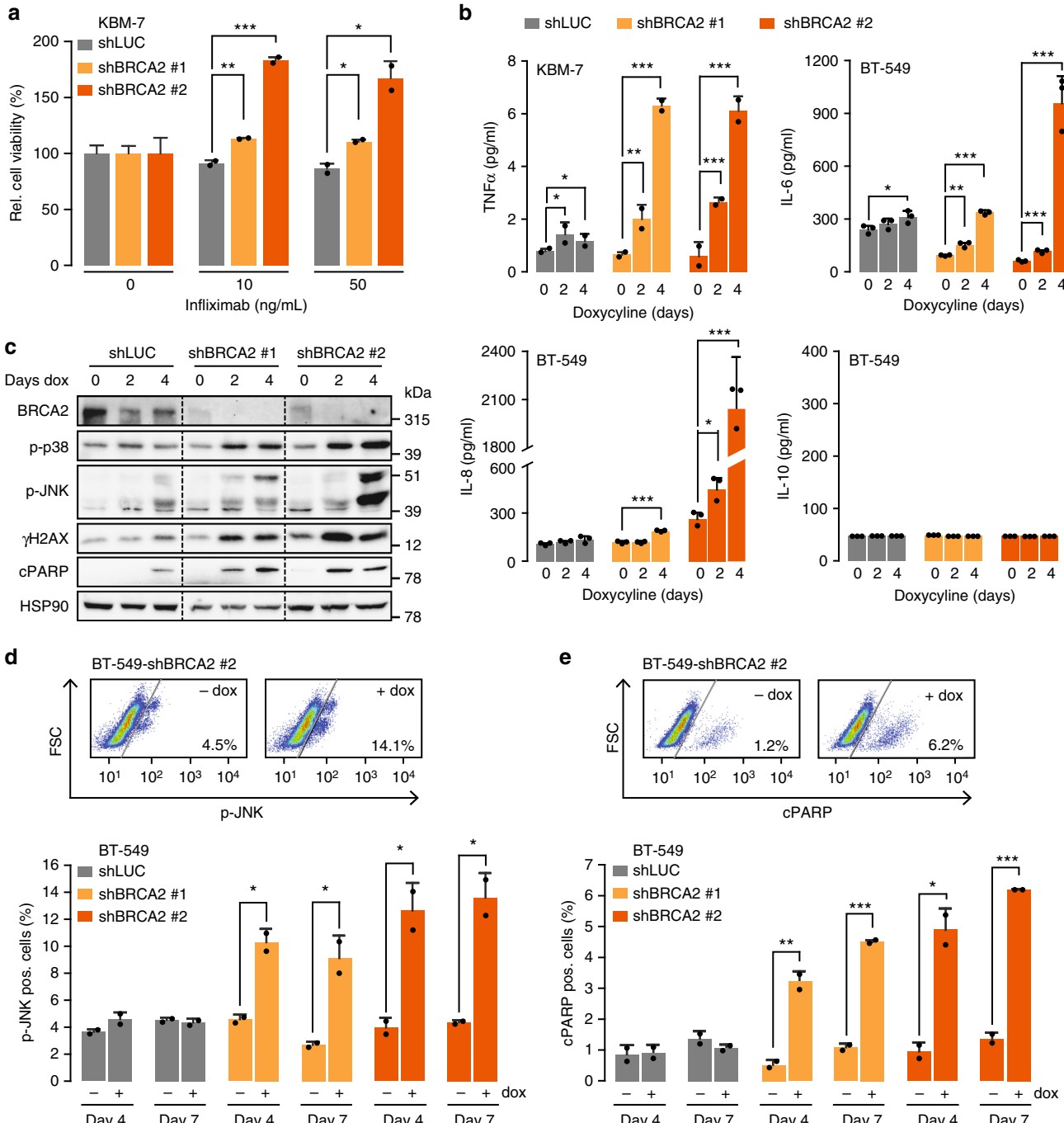

**Fig. 3** BRCA2 depletion results in increased TNFα signaling. **a** KBM-7 cells harboring indicated shRNAs were treated with doxycycline for 48 h and subsequently plated and treated with indicated concentrations of infliximab for 5 days. Error bars indicate s.d. of two independent experiments. **b** Levels of TNFα, IL-6, IL-8 and IL-10 secretion upon BRCA2 depletion. After 0, 2 or 4 days of doxycycline treatment, medium was harvested. TNFα was measured using ELISA. IL-6, IL-8 and IL-10 were measured using bead-arrays. Error bars indicate s.d. of two or three independent experiments. **c** Immunoblotting of BT-549 cells harboring indicated shRNAs after 0, 2 or 4 days of doxycycline treatment. Levels of p-p38, p-JNK, γH2AX, cleaved PARP ('cPARP') and HSP90 were analyzed. Both JNK isoforms (p46 and p54) are indicated. Dotted lines are used to indicate different shRNAs. **d** BT-549 cells harboring indicated shRNAs were treated with doxycycline for the indicated time periods, and analyzed by flow cytometry for p-JNK expression. Gating was performed as shown in the top panel. Numbers indicate the percentages of living cells stained positive for p-JNK. Error bars indicate s.d. of two independent experiments. **e** BT-549 cells were treated as described in **d**. Cells were analyzed by flow cytometry for cleaved PARP. Gating was performed as shown in the top panel. Numbers indicate the percentages of living cells stained positive for cleaved PARP. Error bars indicate s.d. of two independent experiments. Throughout the figure, P values were calculated using two-tailed Student's t-test. *P < 0.05, **P < 0.01, ***P < 0.001

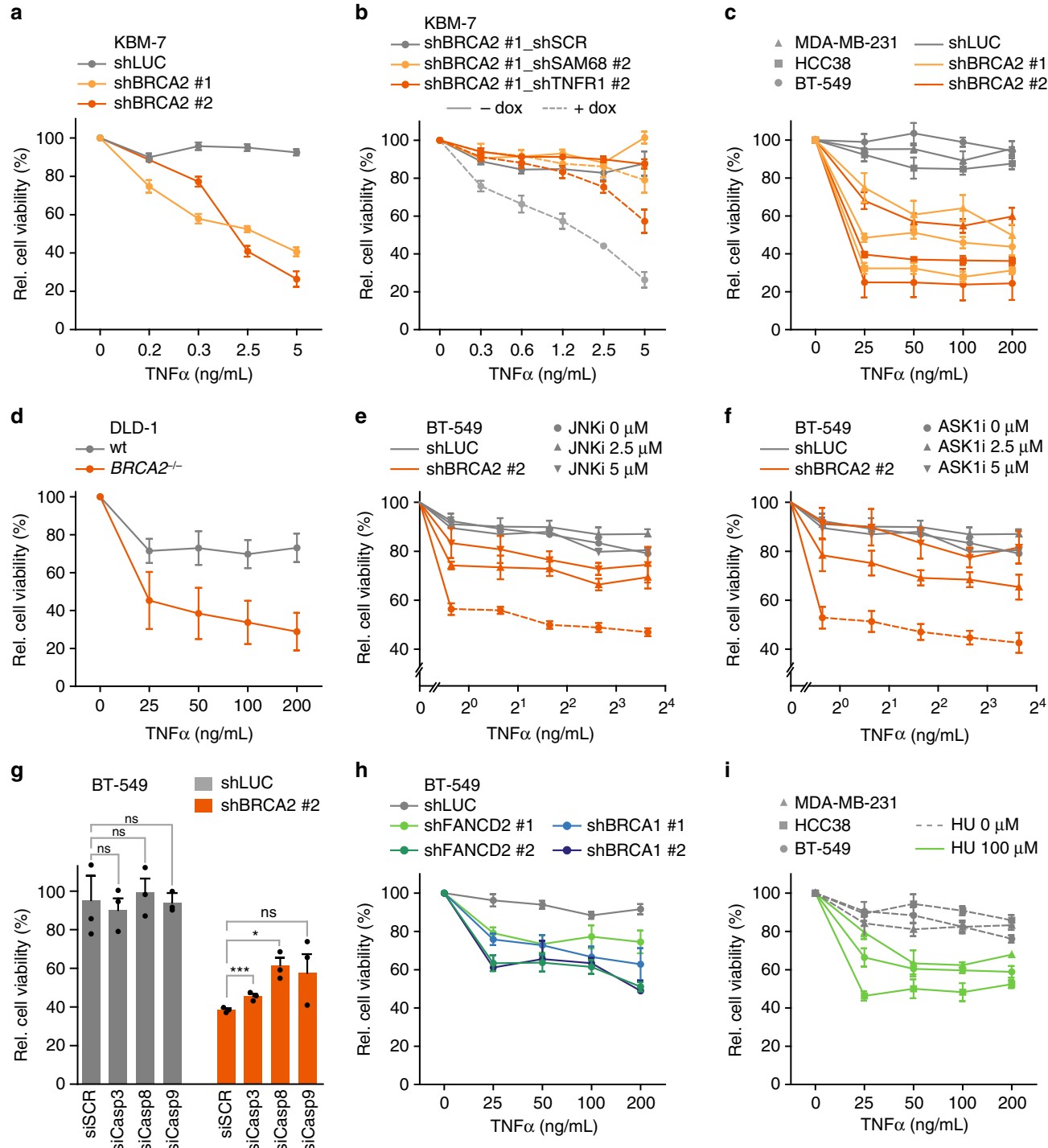

TNFα, and their viability was not affected by JNK or ASK1 inhibition (Fig. 4e, f, Supplementary Fig. 5f). In contrast, BRCA2-depleted cells again showed decreased viability upon TNFα administration. Notably, these effects were dose-dependently reversed by JNK or ASK1 inhibition (Fig. 4e, f, Supplementary Fig. 5f). To test if the observed increase in TNFα sensitivity upon BRCA2 depletion was driven by apoptosis-mediated cell death, we depleted caspase-3, -8 or -9 using siRNAs (Supplementary Fig. 5g). Especially depletion of caspase-8 and -9 resulted in reduced sensitivity to TNFα in BRCA2-depleted cells, while the viability of control-depleted cells was not significantly affected (Fig. 4g, Supplementary Fig. 5h). Blocking caspase activity using the broad-spectrum caspase inhibitor zVAD-FMK confirmed the

requirement of caspase activity, as TNFα-induced cell death in BRCA2-depleted BT-549 and HCC38 cells was significantly rescued by zVAD-FMK treatment (Supplementary Fig. 5i). Combined, these data point at JNK and ASK1 kinases and caspase-8 and -9 to drive TNFα-induced cell death upon BRCA2 inactivation.

To check whether increased TNFα sensitivity was selectively induced by BRCA2 inactivation, BT-549 and HCC38 cells were depleted for the DNA repair proteins BRCA1 or FANCD2 (Supplementary Fig. 6a,b). Depletion of BRCA1 or FANCD2 decreased long-term survival, comparable to BRCA2 depletion (Supplementary Fig. 6c). Importantly, depletion of BRCA1 or FANCD2 also resulted in sensitivity to recombinant TNFα, both

**Fig. 4** BRCA2 inactivation causes sensitivity to TNFα in cancer cells. **a** KBM-7 harboring shRNAs targeting BRCA2 were treated with doxycycline for 48 h and subsequently plated and treated with indicated TNFα concentrations for 5 days. **b** KBM-7-shBRCA2 #1 cells with shRNAs targeting SAM68, TNFR1 or SCR were treated with or without doxycycline and treated with indicated TNFα concentrations for 5 days. **c** Breast cancer cell lines MDA-MB-231, HCC38 and BT-549 harboring shLUC, shBRCA2 #1 or shBRCA2 #2 were pre-treated for 48 h with doxycycline and subsequently treated with indicated TNFα concentrations for 5 days. **d** DLD-1 wt or BRCA2$^{-/-}$ cells were plated and treated for 5 days with indicated TNFα concentrations. **e, f** BT-549 cells harboring shLUC or shBRCA2 #2 were treated with doxycycline for 48 h and subsequently treated with indicated concentrations of TNFα, in the presence or absence of JNK inhibitor (**e**) or ASK1 inhibitor (**f**) for 5 days. **g** BT-549 cell lines harboring indicated shRNAs were transfected with indicated siRNAs for 24 h, and were subsequently treated with doxycycline for 48 h. Cells were re-plated and treated with indicated TNFα concentrations for 5 days. Error bars represent s.e.m. of three independent experiments, with three technical replicates each. P values were calculated using two-tailed Student's t-test. *P < 0.05, ***P < 0.001. **h** BT-549 cells harboring shRNAs targeting BRCA1 or FANCD2 were treated with doxycycline for 48 h, and subsequently plated and treated with indicated TNFα concentrations for 5 days. **i** MDA-MB-231, HCC38 or BT-549 cells were plated and treated with or without 100 μM HU and indicated TNFα concentrations for 5 days. Throughout the figure, cell viability was assessed by MTT conversion, and error bars indicate s.e.m. of at least three independent experiments with three technical replicates each. Measurements were normalized to untreated cells. P values were calculated using two-tailed Student's t-test. For P values, see Supplementary Data 4

in BT-549 and HCC38 cells (Fig. 4h and Supplementary Fig. 6d). Of note, induction of replication stress with a non-toxic dose of hydroxyurea (HU) (Supplementary Fig. 6e, f) also sensitized TNBC cell lines to recombinant TNFα (Fig. 4i, Supplementary Fig. 6g). Thus, TNFα sensitivity is not specific to BRCA2 inactivation, but is also induced by inactivation of BRCA1 or FANCD2, or chemical induction of replication stress.

**BRCA2 inactivation leads to an interferon response**. To investigate how BRCA2 inactivation underlies differential activation of the TNFα pathway, we assessed global changes in protein abundance using SILAC-MS (Stable Isotope Labeling by Amino acids in Cell culture–mass spectrometry) (Fig. 5a). Labeled ('heavy') or unlabeled ('light') protein extracts from BRCA2-depleted or control-depleted BT-549 or HCC38 cells were mixed and analyzed by mass spectrometry (MS). To control for potential effects of metabolic labeling, label-swap controls were included (Fig. 5a). Common differentially expressed proteins measured in at least three out of four independent MS runs were plotted (Fig. 5b, Supplementary Data 3). Interestingly, depletion of BRCA2 resulted in a common set of upregulated proteins (Fig. 5c). When the top 25 upregulated proteins were analyzed using gene set enrichment, a clear enrichment for interferon-α and interferon-γ pathways was found (Fig. 5c). Because mass spec proteomics only captures a subset of the proteome, we validated these observations using gene expression profiling. To this end, gene set enrichment analysis was performed on RNA-sequencing (RNAseq) data derived from control-depleted or BRCA2-depleted BT-549 or HCC38 cells (Fig. 5d–g). Clearly, the most significantly enriched gene sets in both cell lines included interferon-γ and interferon-α responses, as well as activation of TNFα-responsive pathways (Fig. 5d–g, and Supplementary Fig. 7a,b).

**cGAS/STING-dependent TNFα sensitivity upon BRCA2 loss**. Previous studies from us and others have demonstrated that defective DNA repair can lead to aberrant mitoses and micronuclei[36,37]. Recently, cells harboring micronuclei were shown to express a distinct gene expression profile, characterized by cGAS/STING (cyclic GMP-AMP synthase/stimulator of interferon genes)-dependent interferon signaling[38]. RNAseq analysis of BRCA2-depleted BT-549 and HCC38 cells showed a significant enrichment for this 'interferon-stimulated geneset' (Fig. 6a)[38]. In line with this notion, we observed elevated levels of micronuclei and cGAS-positive micronuclei upon BRCA2 depletion in BT-549 and HCC38 cells (Fig. 6b, c), which was accompanied by elevated levels of phosphorylated signal transducer and activator of transcription 1 (STAT1), a key

mediator of interferon-induced transcription (Fig. 6d, Supplementary Fig. 7c). Importantly, siRNA-mediated depletion of cGAS or STING resulted in reduced levels of STAT1 phosphorylation in BRCA2-depleted BT-549 and HCC38 cells (Fig. 6e, f and Supplementary Fig. 7d,e). Furthermore, depletion of cGAS or STING rescued the sensitivity of TNFα upon BRCA2 inactivation in BT-549 and HCC38 cells (Fig. 6g and Supplementary Fig. 7f). These results were confirmed in BT-549 cells in which cGAS was mutated using CRISPR/Cas9. Specifically, mutation of cGAS rescued long-term viability in BRCA2-depleted cells (Supplementary Fig. 7g,h). Also, TNFα sensitivity and STAT1 phosphorylation upon BRCA2 depletion were rescued in cGAS$^{-/-}$ cells compared to cGAS wt cells (Fig. 6h,i and Supplementary Fig. 7i,j). Combined, these data show that BRCA2 inactivation instigates a cGAS/STING-dependent pro-inflammatory response which enhances TNFα sensitivity (Fig. 7).

**Discussion**
DNA repair defects facilitate genome instability and the ensuing accumulation of cancer-promoting mutations[39]. Indeed, inherited or somatic mutations in DNA repair genes are frequently observed in cancer[1]. Yet, defective DNA repair compromises cellular viability, and it remains incompletely clear how (tumor) cells respond to loss of DNA repair pathways. In this study, we describe a prominent transcriptional interferon response upon BRCA2 inactivation, which can be ascribed to genome instability and ensuing cytoplasmic DNA. This response leads to widespread cellular re-wiring, including enhanced sensitivity to TNFα. This latter feature was the basis on which *TNFRSF1A (*encoding TNFR1*)* and *KHDRBS1* (encoding SAM68) were identified to rescue cell death in BRCA2-depleted KBM-7 cells. Specifically, HR-deficiency instigates the production of pro-inflammatory cytokines, including TNFα, which activates TNFα receptor-mediated cell death. Interference with the cytosolic DNA sensor cGAS/STING, the TNFα-receptor, or its downstream signaling components rescued TNFα-induced cell death in BRCA2-depleted cells.

TNFα signaling has previously been described to context-dependently promote cellular survival or promote apoptosis[34]. We find that TNFα signaling in the context of accumulated DNA damage exerts pro-apoptotic effects, either in the context of defective DNA repair or through HU-induced replication stress. These conditions have in common that they induce micronuclei, which were recently shown to be a source of cytoplasmic DNA[38,40]. cGAS/STING activation was previously described to instigate a cell-intrinsic interferon response, resulting in STAT signaling[41]. Indeed, BRCA2 depletion induced cGAS-positive micronuclei, along with increased levels of phosphorylated

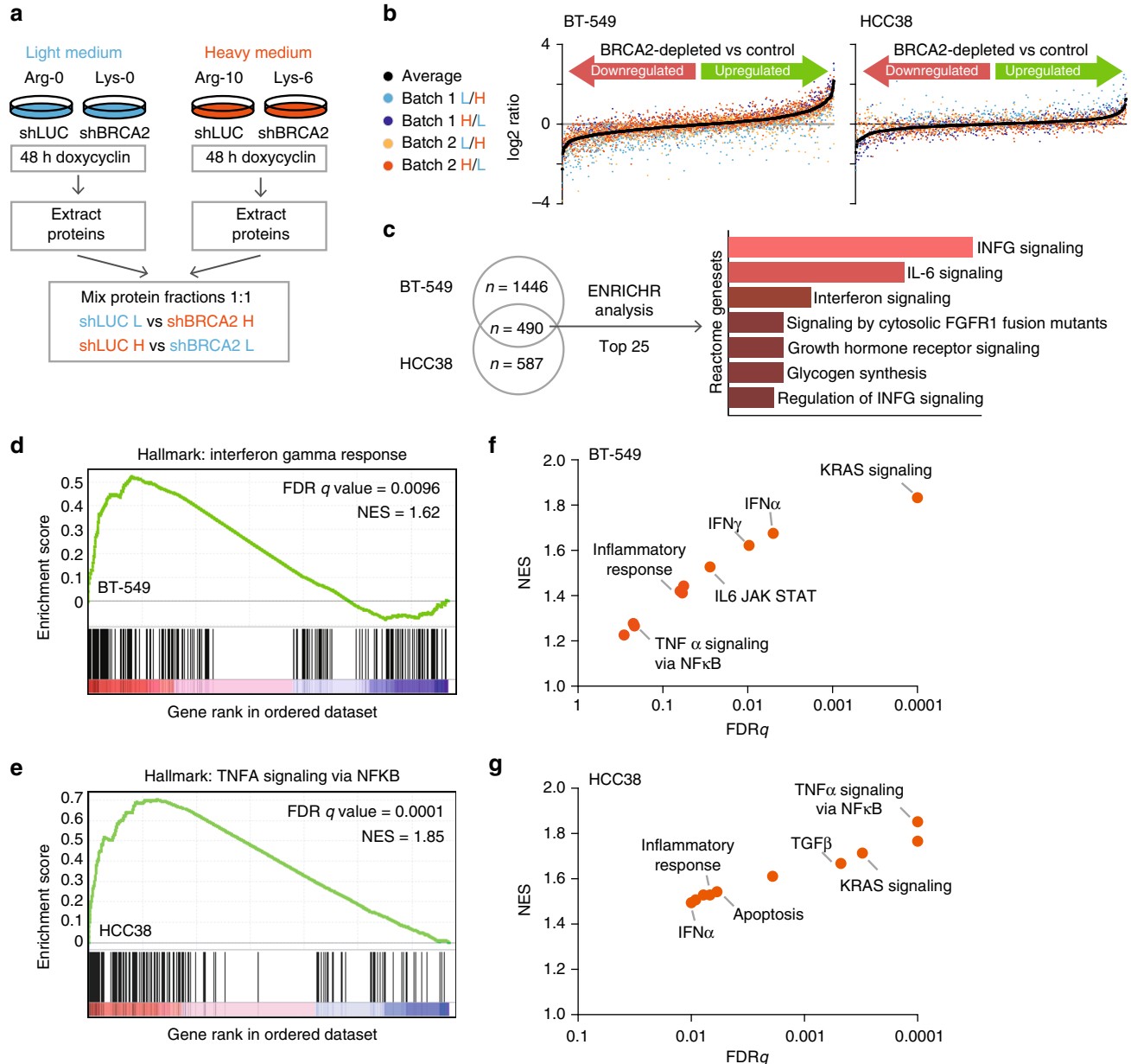

**Fig. 5** Proteomic and transcriptomic analysis reveals upregulation of pro-inflammatory genes upon BRCA2 depletion. **a** Workflow of SILAC-MS analysis of BT-549 and HCC38 cell lines with indicated shRNAs. **b** Log2 ratios (heavy vs light) of proteins that were measured in at least three out of four independent MS analyses in BT-549 (left panel) or HCC38 (right panel) cells. Black dots represent the mean of log2 ratios from three or four experiments. **c** ENRICHR was used to analyze pathway enrichment in top 25 upregulated proteins in response to BRCA2 depletion in BT-549 cells and HCC38 cells. The top 10 enriched Reactome datasets are displayed. **d**, **e** RNA sequencing was performed on BT-549 and HCC38 cells harboring shLUC or shBRCA2 #2, treated for 72 h with or without doxycycline. Gene set enrichment analysis (GSEA) using 'Hallmark' gene sets showed enrichment of Interferon Gamma response (**d**) and TNFA signaling via NF-κB (**e**) in BRCA2-depleted cells. **f**, **g** Top 10 enriched Hallmark gene sets in BRCA2-depleted BT-549 (**f**) and HCC38 (**g**) cells compared to control cell lines. The top 10 list of enriched pathways can be found in Supplementary Fig. 7a

STAT1 (Fig. 6). The observed cellular re-wiring resulted in enhanced TNFα sensitivity, which depended on ASK1 and JNK kinases as well as caspase-8 and -9. Our observation that multiple caspases are involved in TNFα-mediated cell death in BRCA2-defective cells is in line with caspase-8 being engaged in TNFα-mediated apoptosis[42], caspase-9 being involved in intrinsic, DNA damage-induced apoptosis[43] and caspase-3 being a common downstream factor in programmed cell death.

Our findings are also in good agreement with previous reports of increased transcription of TNFα upon irradiation[44], enhanced sensitivity of FANC-C mutant cells to TNFα[45], irradiation-induced re-wiring of TNFα signaling which limited cellular survival[46] and STING activation in response to S-phase DNA damage[47]. Furthermore, treatment with recombinant TNFα was shown to sensitize cancer cells for genotoxic agents[48].

Multiple other mutations have previously been described to rescue cell death upon loss of homologous recombination genes. Most of these mutations (including *TP53BP1*, *MAD2L2*, *HELB* and *RIF1* and Shieldin complex members) could rescue cell death and PARP1 inhibitor sensitivity induced by inactivation of BRCA1 but not BRCA2, which is likely due to BRCA2 functioning downstream of DNA-end resection[23,25,26,49,50]. Recently, inactivation of *PAXIP1*

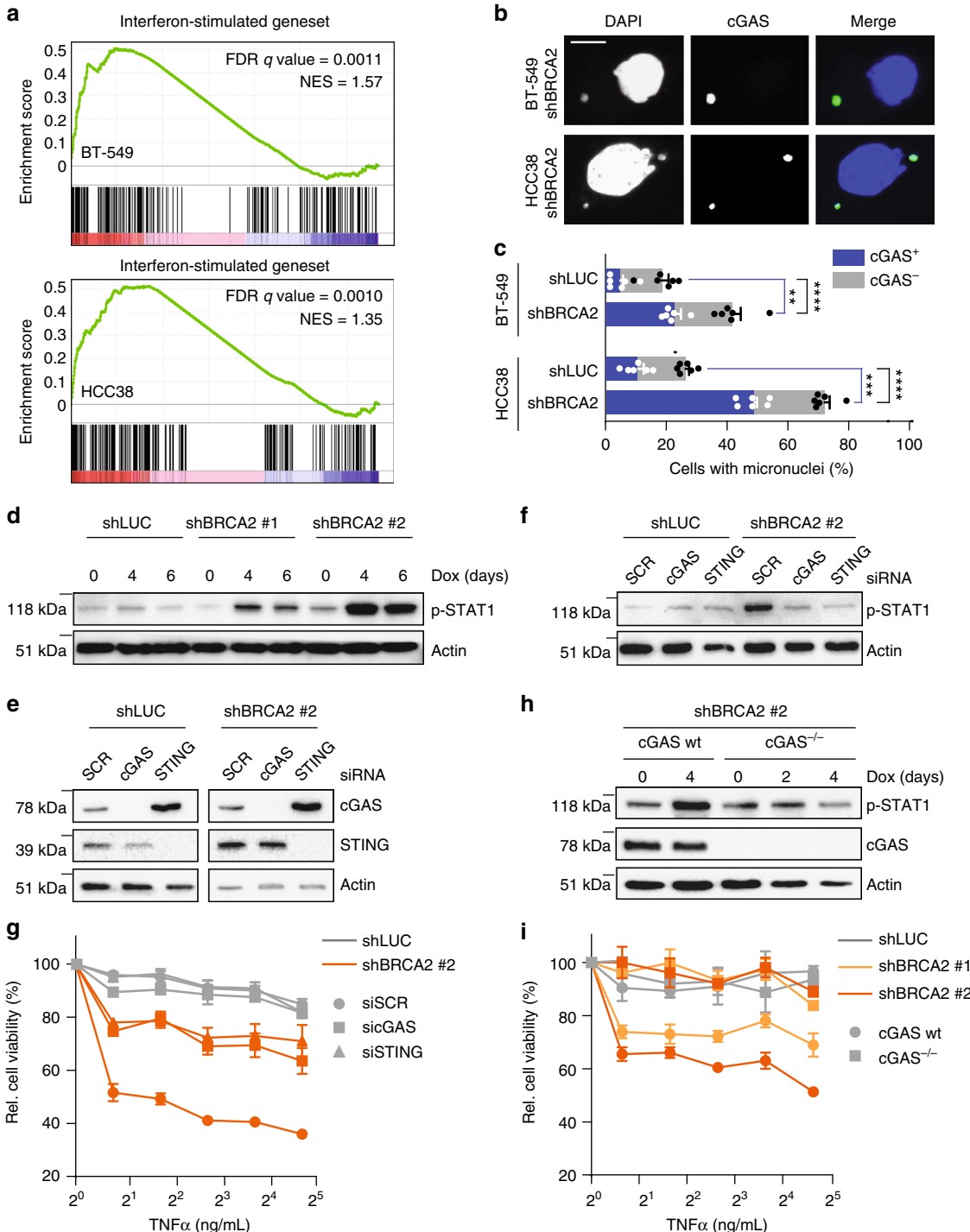

(encoding PTIP) was shown to rescue cell death induced by *BRCA2* mutation[27]. *PAXIP1* was identified in our screen, albeit less significantly enriched when compared to TNFR1 and SAM68.

Constitutive NF-κB activation is described to often occur in different types of cancers, and is associated with aggressive tumor growth and therapy resistance[51]. Recently, and in line with our observations, cancer-associated genomic instability was shown to drive NF-κB activation through a cytosolic DNA response[52]. Such NF-κB activity might be accompanied with autocrine TNFα secretion, as has been demonstrated for head-and-neck cancers[53]. NF-κB activation was previously described in response to DSB formation, where it provides an initial cellular stress response to DNA damage[54,55]. Paradoxically, sustained levels of DNA damage (in our models caused by BRCA2 deficiency) lead to prolonged JNK activation, which is normally suppressed by NF-κB[56,57]. Consequently, sustained JNK signaling can promote pro-apoptotic signaling upon TNFα-induced TNFR1 activation[58,59].

TNFα, in analogy to NF-κB signaling, has also been described to play a role in cancer. Recombinant TNFα was shown to induce cancer cell senescence when combined with interferon-γ treatment, and was demonstrated to induce tumor cell death in metastatic melanoma via isolated limb perfusion[60,61]. Our observations of TNFα sensitivity of BRCA2-defective cancer cells suggest that *BRCA2* mutant tumors may be selectively sensitive to TNFα. Unfortunately, development of TNFα-based treatment modalities was not successful due to toxicity[62]. Conversely, our

**Fig. 6** Micronuclei formation and cGAS/STING-dependent STAT1 activation upon BRCA2 depletion. **a** GSEA shows significant enrichment of interferon-stimulated genes in BRCA2-depleted BT-549 (left panel) or HCC38 (right panel) cells. **b** BT-549 and HCC38 cell lines harboring shLUC or shBRCA2 #2 were treated with doxycycline for 4 days, and stained with anti-cGAS and DAPI. Scale bar represents 15 μm. **c** Quantification of cGAS-positive micronuclei as described in **b**. ≥300 Cells were counted per condition. Error bars indicate s.e.m. of six independent experiments. *P* values were calculated using two-tailed Student's *t*-test. **P < 0.01, ***P < 0.001, ****P < 0.0001. **d** BT-549 cells harboring shLUC or shBRCA2 #2 were treated with doxycycline for indicated time periods. Phosphorylation status of STAT1 was analyzed by immunoblotting. **e** BT-549 cells harboring shLUC or shBRCA2 #2 were transfected with indicated siRNAs. Levels of cGAS and STING were analyzed by immunoblotting at 5 days post transfection. **f** BT-549 cells harboring shLUC or shBRCA2 cells were transfected with indicated siRNAs for 24 h, and subsequently treated with doxycycline for 48 h. Phosphorylation status of STAT1 was analyzed by immunoblotting. **g** BT-549 cells harboring shLUC or shBRCA2 cells were transfected with indicated siRNAs for 24 h. Cells were re-plated and treated with doxycycline for 48 h, followed by treatment with indicated TNFα concentrations for 5 days. **h** cGAS$^{-/-}$ or wt BT-549 cells with indicated shRNAs were pre-treated for 48 h with doxycycline and subsequently treated with indicated TNFα concentrations for 5 days. **i** cGAS$^{-/-}$ or wt BT-549 cells with shBRCA2 #2 were treated with doxycycline for indicated time periods. Phosphorylation status of STAT1 and expression of cGAS were analyzed by immunoblotting. For **g**, **h**, cell viability was assessed by MTT conversion. Error bars indicate s.e.m. of at least three independent experiments with three technical replicates each. Statistical analysis is provided in Supplementary Data 4

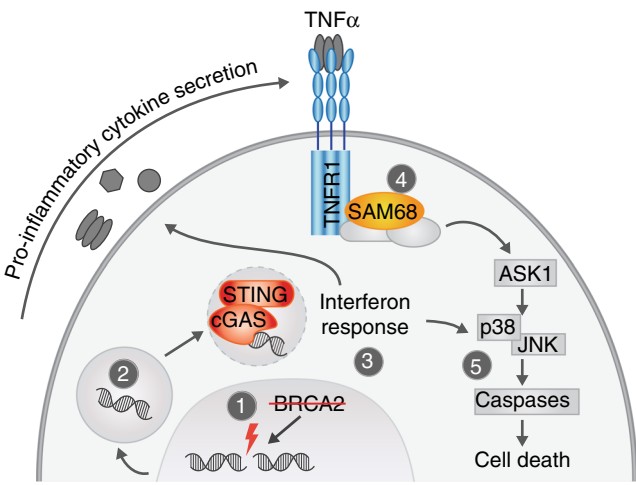

**Fig. 7** Schematic model of inflammatory signaling upon BRCA2 inactivation. BRCA2 inactivation (1) leads to micronuclei formation (2) and cGAS/STING-dependent activation of an interferon response (3). This leads to pro-inflammatory cytokines production, and sensitivity to TNFα, in a TNFR/SAM68 (4) and ASK1/JNK-dependent fashion (5)

data suggest that modulation of TNFα or cGAS/STING signaling may allow survival of BRCA-deficient tumor cells, and warrants care in using TNFα antagonists in *BRCA* mutation carriers.

## Methods

**Cell culture.** KBM-7, BT-549, HCC38, MDA-MB-231 and HEK293T cells were obtained from ATCC. DLD-1 human colorectal adenocarcinoma cells were from Horizon (Cambridge, UK). MEFs harboring the *Brca2*$^{sko}$ allele were a kind gift of Jos Jonkers and Peter Bouwman (Netherlands Cancer Institute, Amsterdam, The Netherlands). Human near-haploid KBM-7 cells were cultured in Iscove's modified Dulbecco's medium. MDA-MB-231 breast cancer cells, 293T human embryonic kidney cells, DLD-1 cells and mouse embryonic fibroblasts were cultured in Dulbecco's modified Eagle's medium. BT-549 and HCC38 were cultured in Roswell Park Memorial Institute (RPMI) medium. Growth media for each line were supplemented with 10% fetal calf serum and penicillin/streptomycin (100 units per mL). All human cell lines were cultured at 37 °C in a humidified incubator supplied with 5% $CO_2$. MEFs were cultured in a low-oxygen (1% $O_2$) incubator. For stable isotope labeling, BT-549 or HCC38 cells harboring shLUC and shBRCA2 #2 were cultured for at least four cell passages (~14 days) in RPMI medium with unmodified arginine (Arg) and Lysine (Lys) (Light 'L') or with stable isotope-labeled Arg[10] and Lys[6] (Heavy 'H') (Silantes).

**Viral transduction.** To generate KBM-7 and breast cancer cell lines with doxycycline-inducible shRNAs, cells were infected with Tet-pLKO-puro, harboring shRNAs directed against luciferase ('shLUC', 5′-AAGAGCTGTTTCTGAGGA GCC-3′), *KIF11* (5′-CACGTACCCTTCATCAAATTT-3′), *BRCA2* (#1 5′-GAA-GAATGCAGGTTTAATA-3′ and #2 5′-AACAACAATTACGAACCAAACTT-3′), *BRCA1* (#1 5′-CCCACCTAATTGTACTGAATT-3′ and #2 5′-GAGTATGCAA ACAGCTATAAT-3′) and *FANCD2* (#1 5′-AAGGGAGGAAGTCATCGAAGT

A-3′ and #2 5′-GGAGATTGATGGTCTACTAGA-3′). Tet-pLKO-puro was a gift from Dmitri Wiederschain (Addgene plasmid #21915)[63].

To validate hits from the genetic screens, KBM-7, MEFs and breast cancer cells were transduced with pLKO.1 vectors, which in addition to the shRNA cassette either carried an IRES mCherry cassette (pLKO.1-mCherry, a kind gift from Jan Jacob Schuringa (UMCG, The Netherlands)) or a puromycin resistance cassette (pLKO.1-puro, a gift from David Root, Addgene plasmid #10878)[64]. Both pLKO.1 plasmids were used as described previously[30]. shRNAs against *TNFRSF1A* and *KHDRBS1* were cloned into pLKO.1 vectors using the *Age*1 and *Eco*R1 restriction sites. The shRNA targeting sequences that were used are: *TNFRSF1A* (#1, 5′-GG AGCTGTTGGTGGGAATATA-3′ and #2, 5′-TCCTGTAGTAACTGTAAGAA A-3′), *KHDRBS1* (#1, 5′-ACCCACAACAGACAAGTAATT-3′ and #2, 5′-GAT GAGGAGAATTACTTGGAT-3′) and SCR (5′-CAACAAGATGAAGAGCACC AA-3′). For MEF cells, shRNA sequences used were for *TNFRSF1A* (5′-GGCTC TGCTGATGGGGATACA-3′), *KHDRBS1* (5′-GACGAGGAGAATTATTTGGA T-3′) and SCR (5′-CAACAAGATGAAGAGCACCAA-3′). Lentiviral particles were produced as described previously[30]. In brief, HEK293T packaging cells were transfected with 4 μg pLKO.1 DNA in combination with the packaging plasmids lenti-VSV-G and lenti-ΔYPR using a standard calcium phosphate protocol. Virus-containing supernatant was harvested at 48 and 72 h after transfection and filtered through a 0.45 μM syringe filter with the addition of 4 μg per mL polybrene. Supernatants were used to infect target cells in three consecutive 12 h periods.

MEFs were transduced with pRetroSuper retrovirus as described previously[23]. Briefly, HEK293T cells were grown to 70% confluency and transfected with 10 μg retroviral vector encoding 'Hit-and-run' Cre recombinase together with Gag-Pol packaging and VSV-G[65]. Supernatants were harvested at 48 and 72 h after transfection and filtered through a 0.45 μM syringe filter. MEFs were plated and infected for 24 h with retroviral supernatant with an additional second and third round of infection after 24 and 32 h. At 24 h after the last infection, cells were washed and cultured in fresh medium with puromycin (2 μg per mL) for 48 h. Switching of the conditional sko allele upon Cre retrovirus, resulting in a 110 base-pair fragment, was shown by PCR amplification of genomic DNA with the following primers: 5′-GTG GGC TTG TAC TCG GTC AT-3′ (forward) and 5′-GTA ACC TCT GCC GTT CAG GA-3′ (reverse).

**Generation of cGAS knockout cells by the CRISPR/Cas9 system.** CRISPR guide RNAs were generated against cGAS (#1 caccgGGCATTCCGTGCG-GAAGCCT; #2 caccgTGAAACGGATTCTTCTTTCG) and cloned into the Cas9 plasmid using the *Age*I and *Eco*RI restriction sites. The pSpCas9(BB)−2A-Puro V2.0 (PX459) was a gift from Feng Zhang (Addgene plasmid #62988)[66]. BT-549 cells were transfected with both guide RNA plasmids simultaneously (2 μg) using FuGene (Promega) according to the manufacturer's instructions. After transfection, cells were selected with puromycin (1 μg per mL) for 2–3 days. Single cell cGAS$^{-/-}$ clones were confirmed by immunoblotting. Subsequently, cGAS$^{-/-}$ or parental cells were infected with Tet-pLKO-puro shRNAs targeting BRCA2 or Luciferase as described before.

**Gene-trap mutagenesis and mapping of insertion sites.** KBM-7 cells were infected with pLKO.1-tet-puro-BRCA2 #2 and puromycin-resistant clones were sorted into monoclonal cell lines. The resulting monoclonal KBM-7-shBRCA2 #2 cell line was mutagenized using retroviral infection as described previously[67]. In short, approximately 64 × 10E6 KBM-7 cells were retrovirally infected with the gene-trap vector pGT, containing a strong splice acceptor. After three consecutive rounds of infection, an ~75% infection rate was achieved based on green fluorescent protein positivity. All mutagenized cells were pooled and 20 × 10E6 cells were treated with 1 μg per mL doxycycline. At 5 days after doxycycline addition, cells were plated at 20,000 cells per well in 40 96-well plates to allow competitive selection for 14 days. Subsequently, cell pellets were frozen and DNA was isolated. Viral insertions were amplified using LAN PCR, identified by massive parallel

sequencing and mapped to the human genome as described previously[68]. DNA sequencing data are available at the NCBI short read archive (PRJNA299537).

**Western blotting**. Knockdown efficiencies and biochemical responses were analyzed by western blotting. Cells were lysed in Mammalian Protein Extraction Reagent (MPER, Thermo Scientific), supplemented with protease inhibitor and phosphatase inhibitor cocktail (Thermo Scientific). Separated proteins were transferred to polyvinylidene fluoride membranes and blocked in 5% milk in Tris-buffered saline, with 0.05% Tween-20. Immunodetection was done with antibodies directed against BRCA2 (1:1000, Calbiochem, #OP95), TNFR1 (1:500, Cell Signaling, #3736; 1:1000, Santa Cruz, sc-8436), SAM68 (1:1000, Santa Cruz, sc-333), BRCA1 (1:1000, Cell Signaling, #9010), FANCD2 (1:200, Santa Cruz, sc-20022), phospho-JNK (1:1000, Cell Signaling, #9251), phospho-p38 (1:1000, Cell Signaling, #4511), cleaved PARP (1:1000, Cell Signaling, #5625), γH2AX (1:1000, Cell Signaling, #9718), phospho-STAT1 (1:1000, Cell Signaling, #9167, #8826), HSP90 (1:1000, Santa Cruz, #sc-69703), cGAS (1:1000, Cell Signaling, #15102), STING (1:1000, Cell Signaling, #13647), caspase-3 (1:1000, Cell Signaling, #9662), caspase-8 (1:1000, Enzo, #ALX-804-242), caspase-9 (1:1000, Cell Signaling, #9502) and beta-Actin (1:10,000, MP Biochemicals, #69100). Appropriate horseradish peroxidase-conjugated secondary antibodies (1:2500, DAKO) were used and signals were visualized with enhanced chemiluminescence (Lumilight, Roche diagnostics) on a Bio-Rad Bioluminescence device, equipped with Quantity One/Chemidoc XRS software (Bio-Rad). Uncropped versions of all western blots can be found in Supplementary Fig. 8–13.

**Quantitative RT-qPCR**. Cell pellets from KBM-7-shBRCA2 #1, KBM-7-shBRCA2 #2 or KBM-7-shLUC treated with doxycycline (1 µg per mL) for 0 or 4 days were harvested. Total RNA was isolated using the RNeasy Mini Kit (Qiagen) and complementary DNA (cDNA) was synthesized using SuperScript III (Invitrogen) according to the manufacturer's instructions. Quantitative reverse transcription-PCR (RT-PCR) for BRCA2 mRNA expression levels was performed in triplicate using the following oligos: 5′-TTGTTTCTCCGGCTGCAC-3′ (forward) and 5′-CGTATTTGGTGCCACAACTC-3′ (reverse). Glyceraldehyde 3-phosphate dehydrogenase (GAPDH) was used as a reference and experiments were performed on an Applied Biosystems Fast 7500 machine. Alternatively, KBM-7-shBRCA2 #2 cells were pre-treated with doxycycline for 24 h prior to transfection with 40 nM SUPTH3 siRNA (Thermo Scientific; ON-TARGETplus SMART pool #L-019548-00) or 'medium GC duplex' control siRNA (Life Technologies, #12935-300). At 72 h after siRNA transfection, cell pellets were harvested and BRCA2 mRNA levels were determined as above.

**Immunofluorescence microscopy**. KBM-7 cells were left untreated or were irradiated using a Cesium[137] source (CIS international/IBL 637 irradiator, dose rate: 0.01083 Gy per second). After 3 h, cells were washed in phosphate-buffered saline (PBS) and then fixed in 2% paraformaldehyde with 0.1% Triton X-100 in PBS for 30 min at room temperature. Cells were permeabilized in 0.5% Triton X-100 in PBS for 10 min. Subsequently, cells were extensively washed and incubated with PBS containing 0.05% Tween-20 and 4% bovine serum albumin (fraction V) (PBS-Tween-BSA) for 1 h to block nonspecific binding. For micronuclei staining, BT-549 and HCC38 cells were grown on coverslips and treated with doxycycline for 4 days. Cells were fixed in 4% paraformaldehyde for 15 min at room temperature. Subsequently, cells were permeabilized with 0.1% Triton X-100 in PBS for 1 min followed by blocking in 0.05% Tween-20 and 2.5% BSA in PBS for 1 h. Cells were incubated overnight at 4 °C with primary antibodies targeting RAD51 (GeneTex, GTX70230, 1:400), γH2AX (Cell Signaling, #9718, 1:100) or cGAS (Cell Signaling, #15102, 1:200) in PBS–Tween–BSA. Cells were extensively washed and incubated for 1 h with Alexa-conjugated secondary antibodies (1:400) and counterstained with 4′,6-diamidino-2-phenylindole (DAPI). Slides were mounted with ProLong Antifade Mountant (Thermofisher). Images were acquired on a Leica DM-6000RXA fluorescence microscope, equipped with Leica Application Suite software.

**DNA fiber assay**. To assess replication fork protection during replication stress, KBM-7-shBRCA2 #2 cells were pre-treated with doxycycline (1 µg per mL) for 96 h, and then pulse-labeled with chloro-deoxyuridine (CIdU, 50 µM) for 40 min. Subsequently, cells were washed with medium and incubated with HU (2 mM) for 4 h. Cells were lysed on microscopy slides in lysis buffer (0.5% sodium dodecyl sulfate (SDS), 200 mM Tris (pH 7.4), 50 mM EDTA). DNA fibers were spread by tilting the slide and were subsequently air-dried and fixed in methanol/acetic acid (3:1) for 10 min. Fixed DNA spreads were stored for 24 h at 4 °C, and prior to immuno-labeling, spreads were treated with 2.5 M HCl for 1.5 h. CIdU was stained with rat anti-BrdU (1:750, AbD Serotec) for 2 h and slides were further incubated with AlexaFluor 488-conjugated anti-rat IgG (1:500) for 1.5 h. Images were acquired on a Leica DM-6000RXA fluorescence microscope, equipped with Leica Application Suite software. The lengths of CIdU and IdU tracks were measured using ImageJ software.

**Flow cytometry**. To measure changes in the fraction of shRNA-containing mCherry-positive cells, cells were re-plated every 3 or 4 days. At those time points, approximately 25% of the culture was used to measure the percentage of mCherry-

positive cells by flow cytometry, whereas the remaining cells were re-plated for further time points. If indicated, cells were treated with doxycycline (1 µg per mL) or ethanol as a solvent control. At least 10,000 (BT-549) or 30,000 (KBM-7) events were analyzed per sample on an LSR-II (Becton Dickinson). Cells, pre-treated with doxycycline (1 µg per mL) or HU, were harvested at different time points, washed and fixed in ice-cold 70% ethanol. Cells were permeabilized and blocked in PBS–1% BSA–0.05% Tween-20 or with PBS–2% BSA–0.1% Triton for 1 h and stained with rabbit anti-cleaved PARP (1:100, Cell Signaling, #5625), rabbit anti-phospho-SAPK/JNK (Thr183/Tyr185) (1:100, Cell Signaling, #9251), rabbit anti-TNFR1 (1:100, Abcam, #19140) or rabbit anti-γH2AX (1:100, Cell Signaling, #9718) overnight at 4 °C. Samples were subsequently stained with AlexaFluor 488-conjugated goat anti-rabbit secondary antibody (1:400) for 1 h and analyzed on a FACS Calibur (Becton Dickinson). Data were analyzed with FlowJo software.

**Clonogenic survival assays**. BT-549 cells or MEFs were plated in 6-well plates (1000 cells per well) and treated with doxycycline (1 µg per mL) or recombinant TNFα as indicated. MEFs were pre-infected with retroviral 'Hit-and-run' Cre recombinase and selected with puromycin (2 µg per mL). After 14 days, cells were fixed in 4% formaldehyde–PBS and stained with 0.1% crystal violet in H₂O. Clonogenic assays were measured and quantified using an EliSpot reader (Alpha Diagnostics International) with vSpot Spectrum software.

**MTT assays**. KBM-7, MDA-MB-231, BT-549, HCC38 and MEF cells were plated in 96-wells plates (600–1000 cells per well), and pre-treated with or without doxycycline (1 µg per mL) for 2 days. MEFs were pre-infected with retroviral 'Hit-and-run' Cre recombinase and selected with puromycin (2 µg per mL). If indicated, BT-549 and HCC38 cells were transfected with siRNAs for 24–48 h prior to plating cells in 96-well plates. Specifically, cells were transfected with siRNA smartpools (final concentration 100 nM), targeting cGAS (#015607, Dharmacon), STING (#024333, Dharmacon), caspase-3 (#29237, Santa Cruz), caspase-8 (#29930, Santa Cruz), caspase-9 (#29931, Santa Cruz) or a negative control sequence (#12935300, Thermofisher) using oligofectamine (Invitrogen), according to the manufacturer's guidelines. Cells were treated with indicated concentrations of the following agents: Infliximab (Merck, Sharp and Dome), HU (Sigma), ASK1 inhibitor NQDI-1 (Axon Medchem, #2179), JNK inhibitor SP600125 (Selleck Chemicals, #S1460), Pan caspase inhibitor (Z-VAD-FMK, Promega) and/or recombinant TNFα (Thermofisher). After 5 days of treatment, methyl thiazol tetrazolium (MTT) was added to a final concentration of 5 mg per mL for 4 h. Medium was removed and formazan crystals were dissolved in dimethyl sulfoxide (DMSO). The absorbance was measured at 520 nm with a Bio-Rad iMark spectrometer. Cell viability was calculated as the relative value in signal compared to DMSO or untreated cells. Unless mentioned otherwise, statistical significance was tested using two-sided Student's t-tests.

**Cytokine analysis**. To analyze excreted TNFα levels, KBM-7-shBRCA2 or KBM-7-shLUC cells were treated with doxycycline (1 µg per mL) for 48 h. Proteins in supernatant culture media were concentrated using Microcon-30 kDa centrifugal filter units with Ultracel-30 membrane (Millipore). Subsequently, TNFα concentrations were determined using a human TNFα ELISA kit (KHC3011, Life Technologies).

IL-6, IL-8 and IL-10 levels were analyzed using the Human Inflammatory Cytokine Kit (BD Bioscience, #551811), according to the manufacturer's protocol. In short, media were collected from BT-549 cells harboring different shRNAs, after treatment with doxycycline for 0, 2 or 4 days. Media samples (50 µL per sample) were incubated with IL-6, IL-8 and IL-10 capture beads for 3 h at room temperature. After two wash steps, samples were measured on an LSR-II (Becton Dickinson). Data were analyzed using FlowJo software, and cytokine concentrations were calculated using cytokine standards (BD Bioscience).

**In-gel digestion and liquid chromatography/tandem mass spectrometry**. BT-549 cells and HCC38 were cultured in light ('L') or heavy ('H') SILAC media and were treated with doxycycline for 48 h. Cells were harvested and lysed in NP-40 buffer (20 mM Tris pH 7.4, 150 mM NaCl, 0.2% v/v Igepal, 10% glycerol) supplemented with a protease/phosphate inhibitor cocktail (Thermofisher). Protein concentrations were determined using Bradford assay and 50 µg of proteins from shLUC-'L' cells was mixed with shBRCA2 #2-'H' cells and vice versa. Proteins were separated by sodium dodecyl sulfate–polyacrylamide gel electrophoresis (SDS-PAGE). Gel lanes were cut into slices for in-gel digestion. Slices were cut into 1 mm pieces and destained with 100 mM ammonium bicarbonate (ABC) in 50–70% acetonitrile. Reduction (10 mM dithiothreitol in 100 mM ABC) and alkylation (55 mM iodoacetamide in 100 mM ABC) steps were performed to block cysteines. Gel pieces were dehydrated and incubated overnight with 10 ng per µL trypsin (Promega), diluted in 100 mM ABC at 37 °C. Peptides were subsequently extracted with 5% formic acid for 20 min.

Online chromatography of the extracted tryptic peptides was performed using an Ultimate 3000 HPLC system (Thermofisher Scientific), coupled to a Q-Exactive-Plus mass spectrometer with a NanoFlex source (Thermofisher Scientific), equipped with a stainless-steel emitter. Tryptic digests were loaded onto a 5 mm × 300 µm internal diameter (i.d.) trapping micro column packed with PepMAP100, 5 µm particles (Dionex) in 0.1% formic acid at the flow rate of 20 µl per min. After

loading and washing for 3 min, trapped peptides were back-flush eluted onto a 50 cm × 75 µm i.d. nanocolumn, packed with Acclaim C18 PepMAP RSLC, 2 µm particles (Dionex). Column temperature was maintained at 40 °C. Eluents used were 100:0 $H_2O$/acetonitrile (volume/volume (V/V)) with 0.1% formic acid (Eluent A) and 0:100 $H_2O$/acetonitrile (v/v) with 0.1% formic acid (Eluent B). The following mobile phase gradient was delivered at the flow rate of 300 nL per min: 3–50% of solvent B in 90 min; 50–80% B in 1 min; 80% B during 9 min, and back to 1 % B in 1 min and held at 1% A for 19 min which results in a total run time of 120 min. MS data were acquired using a data-dependent acquisition (DDA) top-12 method dynamically choosing the most abundant not-yet-sequenced precursor ions from the survey scans (300–1650 Th) with a dynamic exclusion of 20 s. Survey scans were acquired at a resolution of 70,000 at mass-to-charge ($m/z$) 200 with a maximum inject time of 50 ms or AGC 3E6. DDA was performed via higher energy collisional dissociation fragmentation with a target value of $5 \times 10E4$ ions determined with predictive automatic gain control in centroid mode. Isolation of precursors was performed with a window of 1.6 $m/z$. Resolution for HCD spectra was set to 17,500 at $m/z$ 200 with a maximum ion injection time of 50 ms. Normalized collision energy was set at 28. The S-lens RF level was set at 60 and the capillary temperature was set at 250 °C. Precursor ions with single, unassigned, or six and higher charge states were excluded from fragmentation selection.

**MS data analysis**. Mass spectrometry raw files were processed in MaxQuant (version 1.5.2.8) containing the integrated Andromeda search engine and searched against the human proteome downloaded from the UniProt database (20,197 entries), using a false discovery rate of 0.01 at the protein and peptide level. Multiplicity was set to 1 with Lys6 and Arg10 selected as labels. Carbamidomethyl was set as a fixed modification and oxidation of methionine as a variable modification. Default parameters were used for all other settings. Proteins were excluded based on the criteria 'marked potential contaminant or reverse protein by MaxQuant' and 'only identified by either light or heavy labeled peptide'. For further analysis, log2 of protein ratio's (heavy/light) was calculated. Proteins present in at least 3 out of 4 independent analyses were further analyzed. The full datasets as plotted in Fig. 5b were used to select upregulated proteins upon BRCA2 depletion, compared to control shLUC cells. Proteins that were among the top 25 upregulated proteins in both BT-549 and HCC38 cells were selected. The average level of upregulation in the BT-549 and HCC38 datasets was used for ENRICHR analysis[69].

**Gene expression analysis**. For indicated genes, mRNA expression levels from the Ovarian Serous Cystadenocarcinoma TCGA dataset were retrieved from cBioportal. Only tumors with sequencing and CNA data (316 samples) were used, and were subclassified in 'BRCA2 wildtype' (193 samples) and 'germline BRCA2 mutant' (25 samples). Of the 25 germline BRCA2 carriers, 2 cases were excluded because of an additional BRCA1 mutation, resulting in 23 BRCA2 mutant cases. In the 'BRCA2 wildtype' set, only samples were included that did not harbor any alterations (amplification, deletion, mutation, hyper-methylation, mRNA up- or downregulation) in BRCA1 or BRCA2. Two-sided Student's t-tests were used to test for statistically significant differences in mRNA expression levels between BRCA2 wt and BRCA2 mutant tumors.

Next-generation RNA sequencing was performed to analyze changes in gene expression upon BRCA2 depletion. To this end, BT-549 or HCC38 cells harboring shBRCA2 #2 or shLUC were treated with doxycycline (1 µg per mL) for 72 h. Cells were harvested and frozen at −80 °C, and RNA was isolated using the mirVANA kit (Ambion, AM1561). RNA quality was analyzed on microfluidic sipper chips and detected by fluorescence (LabChip GX, Caliper LifeSciences), and samples with RNA Quality Scores (RQS) > 5 were included for analysis. The QuantSeq RNAseq 3'mRNA kit (Lexogen) was used to generate cDNA libraries for next-generation sequencing. The cDNA library was purified and PCR amplified with Illumina sequencing adapters were sequenced with 65 base-pair reads on a NextSeq 500 sequencer (Illumina), and generated 7.2 to 19.8 million reads per sample. RNA sequencing quality control was assessed by FastQC and Samtools Flagstat software.

**RNA sequencing data analysis**. For gene set enrichment analysis (GSEA), genes in each cell line were ranked based on the –log P value between doxycycline-treated shBRCA2 #2 cells and three control settings (shLUC cells and untreated shBRCA2 #2 cells). Genes enriched in BRCA2-depleted cells were positive and genes enriched in control cells were negative. The ranked gene lists were loaded into GSEA software and tested against a set of 280 interferon-induced genes[70]. Furthermore, gene sets of the Hallmark collection (MSigDB) were loaded into GSEA and analyzed in both cell lines. RNA sequencing data are accessible at the GEO repository, under accession number GSE116943.

**Reporting summary**. Further information on experimental design is available in the Nature Research Reporting Summary linked to this Article.

## Data availability

RNA sequencing data are accessible at the GEO repository, under accession number GSE116943. The mass spectrometry data have been deposited to the ProteomeXchange Consortium via the PRIDE partner repository with the dataset identifier PXD007253 (BT-549) and HCC38. DNA sequencing data are available at the NCBI short read archive (PRJNA299537). Other data from this study are available from the corresponding author upon request.

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

## Acknowledgements

We thank members of the van Vugt and Brummelkamp labs, Elisabeth de Vries, Maaike Vreeswijk and John Martens for helpful discussions. This work was financially supported by grants from the Netherlands Organization for Scientific research (NWO-VIDI #91713334 to M.A.T.M.v.V.), the Dutch Cancer Society/Alpe D'huzes (Grant EMCR2014-7048 to M.A.T.M.v.V.), the European Research Council (ERC CoS Grant 682421 to M.A.T.M.v.V.), the Meerema-de Boer Foundation to A.M.H. We thank Jos Jonkers, Peter Bouwman and Christy Hong for reagents, and Anouk Baars, Wellington Mardoqueu Candido and Marieke Everts for technical assistance.

## Author contributions

A.M.H., M.A.T.M.v.V. and T.R.B. conceived the project. A.M.H. and L.T.J performed the genetic screen. A.M.H. and F.T. performed cell biological and biochemical experiments. S.E.v.G. and R.S.N.F. performed RNAseq analysis. A.M.H., F.T., L.T.J., T.R.B. and M.A.T. M.v.V. analyzed data. A.M.H., F.T. and M.A.T.M.v.V. wrote the manuscript. All authors provided feedback on the manuscript before submission.

## Additional information

**Competing interests:** T.R.B. is co-founder and SAB member of Haplogen GmbH and co-founder and managing director of Scenic Biotech. The remaining authors declare no competing interests.

