## [Peer Review File · Nature Communications]

Reviewers' comments:

Reviewer #1 (Remarks to the Author):

The authors found that inactivation of TNFR1 or SAM68 rescued the cell death induced by BRCA2 inactivation in a p53-mutated near-haploid cancer cell line, KBM-7. This happens in a cell type-specific manner, since shRNA-mediated inactivation of TNFR1 or SAM68 did not significantly rescue cellular survival of *Brca2*^{-/-}, *Tp53*^{-/-} MEFs. In MDA-MB231 cells, TNFR1 depletion was not feasible.

In BT-549 cells, SAM68 depletion was not feasible, but TNFR1 inactivation led to increased survival of BRCA2-depleted BT-549 cells. Interestingly, BRCA2 depletion led to increased TNF α secretion from KBM-7 cells. Furthermore, BRCA2-depleted KBM-7 cells showed increased sensitivity to TNF α and this TNF α sensitivity was rescued by co-depletion of SAM68 or TNFR1. BRCA2-depletion/deletion led to TNF α sensitivity in MDA-MB-231, HCC38, BT-549 and DLD-1 cells as well. In BT-549 cells, FANCD2 or BRCA1 depletion also caused TNF α sensitivity. In BT-549 cells, BRCA2 depletion resulted in upregulation of ITGA6, ANXA4, ANPEP, CD9, and FAM129A and downregulation of TAGLN, HSPB1.

Finally, decreased viability upon TNF α treatment of BRCA2-depleted BT-549 cells and MDA-MB-231 cells was reversed by ASK1 or JNK inhibition.

These findings collectively show that BRCA2 depletion leads to rewired NF κ B/ TNF α signaling and that activity of JNK and ASK1 is required for TNF α -induced cell death in BRCA2-depleted cells.

These findings suggest that TNF α signaling pathway is involved in apoptosis in response to BRCA2 loss in certain situations. This is an interesting concept. However, the studies seem incomplete. There are several concerns the authors should address.

- 1) Figure 1F legend is too concise and it is difficult for me to understand this figure. The authors should explain what the x axis (total insertions) is and what the y axis (sense/total insertions) is.
 - 2) Methods for the TCGA data analyses are not described.
 - 3) Figure 2A: The difference of KHDRBS1 mRNA expression between BRCA2 mutated cancers and BRCA2 wild type cancers is not statistically significant.
 - 4) Page 7 Line 12 " In addition, of all SOC-tumors that are classified by the TCGA as having down-regulated KHDRBS1 mRNA (n=8), 37.5% has a mutation in BRCA2." For this statement, methods are not described. What is the definition of "having down-regulated KHDRBS1 mRNA"? Why are there only 8 tumors with down-regulated KHDRBS1 mRNA?
 - 5) Figure 4E: just one FANCD2 shRNA was used for this experiment. The authors should use at least two independent shRNAs or should do rescue experiments using shRNA resistant FANCD2 cDNA.
 - 6) Figure 4E: two BRCA1 shRNAs were used, but only one of them caused increased sensitivity to TNF α . So, the authors should not conclude that BRCA1 deficiency leads to TNF α sensitivity.
 - 7) Figure 4F lacks controls (HU-untreated cells should be used as controls.)
 - 8) For Figures 4C and 5DE, only one BRCA2 shRNA was used. The authors should use at least two independent shRNAs.
 - 9) For some of the key experiments, just one cell line was used and how generalizable the results are is questionable.
- For example, for Figure 5ABCF, only BT-549 cells were used.
For Figure 4E, only BT-549 cells were used.
Ideally, the authors should test multiple cell lines.

Reviewer #2 (Remarks to the Author):

The authors describe the role of TNF mediated cytotoxicity in BRCA2 deficiency. A genomic screen that the authors performed revealed TNF as a critical mediator cellular survival upon BRCA2 deficiency. They further claim that this enhanced TNF sensitivity was associated with aberrant TNFR1/NF- κ B signaling in BRCA2 depleted cells required ASK1 and JNK signaling.

Although the ideas presented here could be potentially interesting, the study is lacking some critical mechanistic explanations. For example, it is not clear how does TNF signaling affect viability of BRCA2 depleted cells. The authors have never found TRADD, FADD, RIP1, or Caspase-8 in their screens and these proteins are clearly essential for TNF induced cell death. On the other hand, the role of SAM68 is very context dependent and it is not known how does this protein affect TNF cell death signaling. Additionally, factors found in mass spec experiment (figure 5) and listed on page 10 are not well known or established regulators of TNF and NF- κ B signaling. The authors also use caspase-8 inhibitor to show it does not have a role in this cell death while general caspases and apoptosis are implicated. Thus, the authors should down-regulate caspase-8 or caspase-9 or FADD to investigate their role in this cell death. This is particularly important given that TNF signaling primarily activates proliferative inflammatory signaling and cell death only in combination with other factors such as inhibitors of protein translation or NF- κ B activation.

Second, some of the effects appear to be cell type specific and not generally applicable (figure S3). Is it possible that BRCA2-TNF link is differentially regulated in human and murine cells? In addition, TNF sensitivity is not specific to BRCA2 (figure 4 and S5) suggesting that TNF might only partially responsible for reported phenotype of BRCA2 depleted cells.

In figure 3 the authors should show western blots for different levels of pJNK and PARP cleavage and check if NF- κ B, p38 and ERK signaling and protein activation (phosphorylation) have been affected. Also, it is important to evaluate if NF- κ B and MAPK dependent cytokine and chemokine production have been affected.

Point-by-point rebuttal

First, we would like to thank the reviewers for their constructive remarks. We feel that our manuscript has significantly improved, based on their comments.

Reviewer 1:

Comment: “Figure 1F legend is too concise and it is difficult for me to understand this figure. The authors should explain what the x axis (total insertions) is and what the y axis (sense/total insertions) is.”

Reply: We agree that the original figure caption to panel 1F was very concise. We have adjusted the text next to the X and Y axes, and have changed the legend text for Figure 1F into the following:

“Insertions sites identified in gene-trap mutagenized KBM-7 cells which survived doxycycline-induced BRCA2 inactivation (shBRCA2 #2). Dots represent individual genes. The frequency of an insertion site mapped to a specific gene is plotted on the X-axis. The ratio of gene-traps inserted in the sense orientation over total insertions are plotted on the Y-axis. Genes that are neutral in conferring a survival advantage in BRCA2-depleted cells have a sense/total insertion ratio of 0.5 (indicated by the orange dashed line). Insertion site ratios >0.5 represent genes that when mutated confer survival benefit to BRCA2-depleted cells.”

Comment: “Methods for the TCGA data analyses are not described.”

Reply: We have expanded our methods section on page 20 (‘gene expression analysis’), which includes methodology of how the TCGA data was analyzed.

Comment: “Figure 2A: The difference of KHDRBS1 mRNA expression between BRCA2 mutated cancers and BRCA2 wild type cancers is not statistically significant.”

Reply: We agree with the reviewer that the observed difference in KHDRBS1 mRNA levels is not statistically different between BRCA2 mutated cancers and BRCA2 wild type cancers.

We have now changed the text on page 6 (see new text below), and have moved the mRNA analysis to Supplemental Figure S2A.

“Interestingly, the TNF α pathway component KHDRBS1 on average showed lower median mRNA levels in BRCA2 mutated versus BRCA2 wildtype (wt) tumors (Fig. S2A). KHDRBS1 showed a larger difference in expression level when compared to PAXIP1 or TNFR1, although differences for these genes were not statistically significantly, likely due to the low number of BRCA2 mutant cancers.”

Comment: “Page 7 Line 12 “ In addition, of all SOC-tumors that are classified by the TCGA as having down-regulated KHDRBS1 mRNA (n=8), 37.5% has a mutation in BRCA2.” For this statement, methods are not described. What is the definition of “having down-regulated KHDRBS1 mRNA”? Why are there only 8 tumors with down-regulated KHDRBS1 mRNA?”

Reply: In our initial analysis, we used a very stringent cut-off to define tumors with down-regulated KHDRBS1 mRNA. We agree with the reviewer that this number is low. Also, because it is difficult to extrapolate mRNA to SAM68 function, we have decided to remove this statement from the results section.

Comment: “Figure 4E: just one FANCD2 shRNA was used for this experiment. The authors should use at least two independent shRNAs or should do rescue experiments using shRNA resistant FANCD2 cDNA.”

Reply: For the experiments in Figure 4H and supplemental Figures S6B/C/D, we have now included a second FANCD2 shRNA, and have repeated the experiments in HCC38 cells. Importantly, our new data confirm our initial observation that FANCD2 depletion leads to TNF α sensitivity in BT-549 and HCC38 cell lines.

Comment: “Figure 4E: two BRCA1 shRNAs were used, but only one of them caused increased sensitivity to TNF α . So, the authors should not conclude that BRCA1 deficiency leads to TNF α sensitivity.”

Reply: We agree with the reviewer that one of the BRCA1 shRNAs did not show a significantly increased TNF α sensitivity, which we think may be due to knock-down efficiency for that specific shRNA. We have tested additional BRCA1 shRNAs, and have included new experiments with a BRCA1 shRNA that showed greater knockdown efficiency. Using these reagents, we now show in Figure 4H and Supplemental Figure 6A/C/D that BRCA1 depletion leads to TNF α sensitivity, both in BT-549 and HCC38 cells.

Comment: “Figure 4F lacks controls (HU-untreated cells should be used as controls.)”

Reply: We have repeated the experiments including the requested HU-untreated controls. These data are now included in Figure 4I. These new data show that control-treated cells are not sensitive to TNF α .

Comment: “For Figures 4C and 5DE, only one BRCA2 shRNA was used. The authors should use at least two independent shRNAs.”

Reply: In our revised manuscript, we have now included data using a second BRCA2 shRNA. In Figure 4C, a second BRCA2 shRNA is now used for all three cell lines. For Figures 5D/E, experiments have been repeated in a second cell line (HCC38) with two BRCA2 shRNAs (now Figure 4E,F and S5E,F). Importantly, our new data are in good agreement with our initial results, and strengthen our conclusions that BRCA2 depletion results in TNF α sensitivity, which is dependent on ASK1 and JNK. Finally, new data that shows increased expression of proteins involved in TNF α signaling, is performed with two BRCA2 shRNAs (Figure 3C).

Comment: “For some of the key experiments, just one cell line was used and how generalizable the results are is questionable. For example, for Figure 5ABCF, only BT-549 cells were used. For Figure 4E, only BT-549 cells were used. Ideally, the authors should test multiple cell lines..”

Reply: We have now repeated multiple key experiments in other cell lines.

First, we have repeated the mass spec experiments (Fig 5A,B,C) in HCC38 breast cancer cells. Using the combined proteomics data of HCC38 and BT-549 cells, we compiled a list of proteins that were upregulated in BRCA2-depleted cells compared to control cells. Using pathway enrichment analysis, we show that ‘interferon signaling’ and ‘IL-6 signaling’ -among related pathways- show prominent enrichment.

Second, we have confirmed our proteomic data using RNAseq. We have again employed both BT-549 and HCC38 cells, and observed that upon BRCA2 inactivation, gene expression signatures reflecting interferon response and TNF α signaling were highly enriched. Combined, these new data confirm using multiple cell line models that TNF α signaling is re-wired, as part of a prominent interferon response.

Additionally, we have repeated experiments in Figure 5F, testing the involvement of caspases (specifically caspase 3, 8 and 9) in both BT-549 and HCC38 cells (now Figure 4G and S5G,H,I). Finally, the newly added data in Figure 6, in which we show that BRCA2 depletion leads to cGAS-positive micronuclei, cGAS/STING-dependent STAT1 activation and

cGAS/STING-dependent TNF α sensitivity, were performed in BT-549 as well as HCC38 cells (Figure 6 and Figure S7).

Reviewer #2 (Remarks to the Author):

The reviewer raised several points which we extracted and copied from the text and replied to point by point.

‘The authors describe the role of TNF mediated cytotoxicity in BRCA2 deficiency. A genomic screen that the authors performed revealed TNF as a critical mediator cellular survival upon BRCA2 deficiency. They further claim that this enhanced TNF sensitivity was associated with aberrant TNFR1/NF- κ B signaling in BRCA2 depleted cells required ASK1 and JNK signaling.’

Comment: “Although the ideas presented here could be potentially interesting, the study is lacking some critical mechanistic explanations. For example, it is not clear how does TNF signaling affect viability of BRCA2 depleted cells.”

Reply: We have performed several additional experiments to mechanistically explain how BRCA2 inactivation leads to TNF α sensitivity. Firstly, we have expanded our mass spec analysis and additionally performed RNAseq analysis, which show that BRCA2 depletion leads to a prominent interferon response, both in BT-549 and HCC38 cells (added in Figure 5). The observed gene set enrichment showed striking overlap with a recent RNAseq analysis of micronucleated cells (Mackenzie et al, *Nature* 2017, PMID: 28738408)(newly added Fig. 6A). Indeed, when we analyzed BRCA2-depleted cells, we observed a significant increase in the amount micronuclei (newly added Fig. 6C). These micronuclei frequently recruited the cytoplasmic DNA sensor cGAS, which in conjunction with STING is known to trigger interferon signaling through STAT1. We now show in both BT-549 and HCC38 cells that BRCA2 depletion leads to STAT1 phosphorylation in a cGAS/STING-dependent fashion (now included in Figure 6D,E and Supplemental Figure 7C,D). Finally, we show that the TNF α sensitivity upon BRCA2 inactivation is largely rescued when cGAS or STING is depleted (Fig. 6F and Suppl. Fig. 7E).

Comment: “The authors have never found TRADD, FADD, RIP1, or Caspase-8 in their screens and these proteins are clearly essential for TNF induced cell death.”

Reply: We agree that TRADD, FADD, RIP1 and Caspase-8 are known factors in TNF-induced cell death. Caspase-8 was actually significantly enriched in our screen (ranked #64, Supplemental Table 2). We have now included Caspase-8 in the screen plot in Figure 1. The role of Caspase-8 was further confirmed by new data, in which Caspase-8 depletion was shown to result in a partial rescue in viability both in BT-549 cells (Figure 4G and Supplemental Figure S5G) and HCC-38 cells (Supplemental Figure 5G, H).

Concerning TRADD, FADD and RIP1, these genes were not identified as statistically significantly hits in our screen. In case of TRADD and FADD, the absence of these genes is due to relatively low numbers of mapped integration sites (6 and 4 insertion sites respectively, Supplemental Table 1). RIPK1 is likely essential in KBM-7 cells, as only insertions in reverse orientation were identified (Supplemental Table 1).

Comment: “On the other hand, the role of SAM68 is very context dependent and it is not known how does this protein affect TNF cell death signaling. Additionally, factors found in mass

spec experiment (figure 5) and listed on page 10 are not well know or established regulators of TNF and NF- κ B signaling.”

Reply: We agree with the reviewer that the role of SAM68 in TNF-mediated cell death signaling is not entirely clear, although a role for SAM68 in this pathway has been demonstrated previously (Ramakrishnan and Baltimore, *Mol Cell* 2011, PMID: 21620750). Our results indeed show that SAM68 is context-dependent, and we address this in the discussion section. Importantly, in cell lines in which SMA68 is not essential for viability, depletion of SAM68 rescues TNF α sensitivity.

We also agree with the reviewer that the factors found in the mass-spec are not among the best-described regulators of TNF and NF- κ B signaling. In our revised manuscript, we have no longer high-lighted these proteins in our mass spec plot. Instead, we repeated the mass-spec analysis in control versus BRCA2-depleted HCC-38 cells, and performed gene-set enrichment analysis on those proteins that were quantitatively assessed in datasets from BT-549 and HCC38 (now included in Figure 5A,B,C). Using ENRICH analysis, these mass-spec datasets show a clear and statistically significant enrichment for interferon and IL-6 signaling components.

Although SILAC proteomics provides a solid fingerprint of pathway re-wiring, only a subset of all gene products are quantitatively measured. To this end, we included RNAseq analysis. These data is now included in Figure 5D,E. Again, we observed very significant enrichment for gene sets related to TNF α -signaling through NF- κ B and interferon signaling.

Comment: *“The authors also use caspase-8 inhibitor to show it does not have a role in this cell death while general caspases and apoptosis are implicated. Thus, the authors should down-regulate caspase-8 or caspase-9 or FADD to investigate their role in this cell death. This is particularly important given that TNF signaling primarily activates proliferative inflammatory signaling and cell death only in combination with other factors such as inhibitors of protein translation or NF- κ B activation.”*

Reply: We thank the review for this suggestion. We agree that down-regulation of caspases is more specific when compared to caspase inhibitors. We have performed experiments in which caspase 3, 8 and 9 are depleted using siRNA, both in BT-549 and HCC38 cells. The result from these experiments, which are now included in Figure 4G and Supplemental Figure S5G/H confirm the requirement of caspase-8 and 9, whereas caspase-3 depletion only showed minimal effects.

Comment: *“Second, some of the effects appear to be cell type specific and not generally applicable (figure S3). Is it possible that BRCA2-TNF link is differentially regulated in human and murine cells? In addition, TNF sensitivity is not specific to BRCA2 (figure 4 and S5) suggesting that TNF might only partially responsible for reported phenotype of BRCA2 depleted cells.”*

Reply: Our new data suggests that BRCA2 depletion results in a strong interferon response, which due to a cGAS/STING dependent cytoplasmic DNA response. Cytoplasmic DNA can originate from micronuclei, which are vulnerable structures, and easily leak DNA into the cytoplasm. These observations also explain why depletion of BRCA1 or FANCD2 or HU treatment give similar results, as these conditions were previously also shown to cause micronucleation. We think that the increased TNF-sensitivity is one aspect of the interferon response, and it is this feature that we picked up in our initial genetic screen.

Concerning the generic nature of our finding, we found that HR-deficiency or HU-mediated -replication stress induced TNF-sensitivity in all tested human models, and our proteomic and transcriptomic analysis showed a great overlap between BT-549 and HCC38 cells.

Yet, we do agree with the reviewer that there is context dependence. However, the context dependence largely lies in the degree of essentiality of TNFR and SAM68.

We also agree there is a clear difference of the TNF-sensitivity of the human cell lines that we tested and the MEFs. For transparency, we included these data in our manuscript. As the reviewer suggests, it might be that the interferon response and TNF-sensitivity is differently regulated between human and mouse cells, although the MEFs only represent one murine model. We have addressed these points in our updated discussion section.

Comment: *“In figure 3 the authors should show western blots for different levels of pJNK and PARP cleavage and check if NF- κ B, p38 and ERK signaling and protein activation (phosphorylation) have been affected.”*

Reply: In the newly added Figure 3C, we have included immunoblots of phospho-JNK and phospho-p38, cleaved-PARP and γ H2AX. Clearly, levels of all these markers were upregulated in response to BRCA2 inactivation.

Comment: *“Also, it is important to evaluate if NF- κ B and MAPK dependent cytokine and chemokine production have been affected.”*

Reply: We have now included analysis of a broader cytokine panel. Besides TNF α , BRCA2 inactivation also resulted in production of pro-inflammatory cytokines IL-6 and IL-8. In contrast, we did not observe increased release of the anti-inflammatory cytokine IL-10.

These new data are now included in Figure 3B. These data underscore the broad interferon response that we observed in our RNAseq analysis (now included in Figure 5D,E,F,G and Figure 6A).

Reviewers' comments:

Reviewer #1 (Remarks to the Author):

The authors have addressed all of my concerns, and the manuscript has been improved.

Minor points.

- 1) Typo: P. 3 L. 2. DNA damage Response DNA damage response
- 2) Typo: P. 6 L.13. statistically significantly statistically significant

Reviewer #2 (Remarks to the Author):

In the revised version of their manuscript the authors have attempted to answer reviewers' criticism and were partially but not completely successful.

The issue of different shRNAs for BRCA1 and 2 is not fully answered as it was shown only in some but not all cells. The verification of the knockdown by western blotting is also shown in just few panels but it should be shown in every group of experiments. Importantly, different shRNAs for BRCA2 often show very different effects on signaling; for example differential pJNK activation in figure 3C, or pSTAT1 in figure 6D. These data suggest off-target effect of at least one of these reagents or the effect of clonality. Which one is it?

It is still not clear how SAM68 effects viability in BRCA2 knockdown cells. It is very puzzling that knockdown of SAM68 provides better protection in KBM7 cells from TNF mediated cytotoxicity than knockdown of TNFR1, a critical receptor for TNF. What is the explanation in figure 4B? The authors should show the levels of SAM68 and TNFR1 knockdown by western blotting as it is possible/likely the knockdown was not efficient?

The effect of caspase-8 and caspase-9 knockdown is small (around 20%) but the fact that knockdown of each of them shows similar effect is not consistent with TNF induced cell death which requires caspase-8 but not caspase-9. So why does caspase-9 knockdown have such an effect? Caspase-3 is in general required for all forms of apoptosis but maybe these cells have high levels of caspase-7 and rely on caspase-7 more than caspase-3. Or maybe this is not just apoptosis but there is a strong necrosis/necroptosis component, especially given the partial effect of zVAD treatment. These issues should be thoroughly examined.

Point-by-point rebuttal

We would like to thank the reviewers again for their time and efforts to provide constructive remarks to our revised manuscript.

Reviewer 1:

We are pleased to read that the reviewer is now satisfied with our revised manuscript. We apologize for the two typos. We have corrected both in our newly revised manuscript.

Reviewer 2:

We are pleased to read that the reviewer is satisfied with most of the changes and newly added data in our revised manuscript. The reviewer mentions three additional comments.

Comment 1: *The reviewer asks what the underlying reason could be of the differential effects of shBRCA#1 and shBRCA#2. Specifically, the reviewer asks whether these effects are due to of target effects or effects of clonality.*

Reply:

Our data consistently show similar effects of shBRCA#1 and shBRCA2#2. We do agree with the reviewer that the effect size differs. Specifically, throughout our experiments, shBRCA2#2 gives a stronger phenotype.

Concerning off-target effects, we cannot fully exclude that our BRCA2 hairpins do not have any off-target effects. Yet, we consistently observed similar phenotypes with two independent hairpins, and TNF α sensitivity was confirmed in BRCA2-knockout cell lines.

The difference in effect size between shBRCA#1 and shBRCA2#2 may be attributed to small differences in knockdown efficiency. We have now added western blots for experiments in Figure 3C and 6D (now included in Supplemental Figure 7C). Although effective BRCA2 knock-down was observed with both BRCA2 shRNAs at multiple time-points, western blot analysis for BRCA2 is generally inefficient, with a consequent low dynamic range. This makes it difficult to observe and quantify the small differences in knockdown efficiency that may underlie biological differences.

Although not requested by the reviewer, in our revised manuscript we now included new data, in which we confirmed the involvement of cGAS using a cGAS knockout BT-549 cell line (Figure 6H,I and Supplemental Figure S7G-J).

Comment 2: *“It is still not clear how SAM68 effects viability in BRCA2 knockdown cells. It is very puzzling that knockdown of SAM68 provides better protection in KBM7 cells from TNF mediated cytotoxicity than knockdown of TNFR1, a critical receptor for TNF. What is the explanation in figure 4B? The authors should show the levels of SAM68 and TNFR1 knockdown by western blotting as it is possible/likely the knockdown was not efficient?”*

Reply:

As requested by the reviewer, we have now added western blots for the experiment in Figure 4B (added in Supplemental Figure 5A). In line with western blots in Supplemental Figure 2A, we observe that shRNA#1 for SAM68 consistently shows

somewhat lower knockdown efficiency in comparison with TNFR1 shRNA#1 in KBM-7 cells. These differences in knockdown efficiency may explain the extent to which these shRNAs rescue BRCA2-depleted KBM-7 cells.

Comment 3: “the effect of caspase-8 and caspase-9 knockdown is small (around 20%), but the fact of each of them show similar effect is not consistent with TNF induced cell death.Maybe these cells have high levels of caspase-7 and rely on caspase-7 more than caspase-3?...”

Reply:

We agree with the reviewer that caspase-8 or caspase-9 depletion does not result in a full rescue of the TNF α -induced cell death, which could be attributed to incomplete knock-efficiencies.

Concerning the suggestion that high levels of caspase-7 may underlie our effects, we have analyzed the expression levels of various caspases. We retrieved the RNA sequencing data of the Cancer Cell Line Encyclopedia (Broad Institute, Cambridge, MA, USA) and plotted the relative levels of caspase-3, caspase-7, caspase-8 and caspase-9 (Rebuttal Figure 1A). Analysis of caspase expression levels in BT-549 and HCC-38 showed that these cell lines do not display deviating levels. Specifically, BT-549 and HCC-38 showed lower than median expression levels among breast cancer cell lines (Rebuttal Figure 1A).

Also when we analyzed the expression of caspase-7 in response to BRCA2 depletion, we did not see significant differences in caspase-7 expression (Rebuttal Figure 1B, C).

Our observation that both caspase-8 and caspase-9 are involved in TNF α -induced cell death in BRCA2-depleted cells is actually in line with various observations in literature. Multiple studies have shown that TNF α -induced apoptosis depends on caspase-8, in accordance with the reviewer’s comment. For instance, caspase-8 deletion in mice renders cells insensitive to TNF α (Varfolomeev et al, *Immunity* 1999. PMID: 9729047).

Our data show that TNF α alone hardly induces cell death in the tested cell line models. Only when BRCA2, BRCA1 or FANCD2 are inactivation or when cells are treated with hydroxyurea, cells become sensitive to TNF α -induced apoptosis. In such situations of DNA damage, caspase-9 was shown to become activated and cleaves caspase 3 to stimulate apoptosis (e.g. Ochs and Kaina, *Cancer Research* 2000. PMID 11059778).

Combined, our data show that a cell intrinsic cue (inactivation of DNA repair, or induction of DNA repair) in combination with an extrinsic cue (TNF α) provokes cell death. This notion is in line with a requirement for caspases involved in intrinsic apoptosis (i.e. caspase-9) as well as caspases involved in TNF α -induced extrinsic apoptosis. We have now addressed this issue in the discussion section of our revised manuscript on page 11.

Rebuttal Figure 1

A

[Redacted]

	BT-549 ●	HCC38 ●
Caspase-3	4.446	4.350
Caspase-7	2.079	2.832
Caspase-8	1.996	2.223
Caspase-9	0.996	1.445

Figure legend:

A. RNA sequencing data for indicated caspase genes were retrieved from the Cancer Cell Line Encyclopedia (CCLE, <https://portals.broadinstitute.org/ccle>). RNAseq values for BT-549 and HCC38 are indicated.

B, C. RNA sequencing dataset from Figure 5D-G was analyzed for expression of Caspase-7 in BT-549 (panel B) and HCC38 (panel C) cell lines upon BRCA2 depletion. No statistically significant difference in caspase-7 expression was observed between control cells and BRCA2-depleted cells.